

Water chemistry and greenhouse gas concentrations in waterbodies of a thawing permafrost
peatland complex in northern Norway
Jacqueline K. Knutson[1], François Clayer[1], Peter Dörsch[2,3], Sebastian Westermann[2,4], Heleen
A. de Wit[1,2].
1 Norwegian Institute for Water Research, Økernveien 94, 0579, Oslo, Norway
2 Centre for Biogeochemistry in the Anthropocene, University of Oslo, Oslo, 0371, Norway
3 Faculty of Environmental Sciences and Natural Resource Management, Norwegian
University of Life Sciences (NMBU), 1433 Ås, Norway
4 Department of Geosciences, University of Oslo, Oslo, 0371, Norway
Correspondence: Jacqueline K. Knutson (jacqueline.knutson@niva.no)



## Abstract

Thermokarst ponds in thawing permafrost landscapes play a considerable role in greenhouse
gas (GHG) emissions despite their small size, yet they remain underrepresented in Earth
system models. At the Iškoras site in northern Norway, a peat plateau with decaying
permafrost and thermokarst ponds adjacent to a wetland, we studied water chemistry,
dissolved organic matter (DOM) processing, and GHG fluxes over two years. Thermokarst
ponds exhibited low pH, high organic acidity, and high oversaturation of dissolved carbon
dioxide ($CO_2$) and especially high dissolved methane ($CH_4$). Adjacent wetland streams,
however, with near-neutral pH, showed lower $CH_4$ and organic acidity but significantly
higher $CO_2$ emissions despite moderate saturations driven by turbulence and bicarbonate
replenishment. By contrast, $CO_2$ emissions in ponds were primarily linked to DOM
mineralization.

DOM mineralization rates were similar between ponds and streams, suggesting that
environmental factors like pH and microbial community differences counteract DOM lability
variations. As permafrost decays and transitions from peat plateaus to wetlands, ponds as
hotspots of CH4 emissions will disappear. However, total GHG fluxes across the peatland-
wetland continuum will depend on wetland emissions, where CH4 emissions usually are
considerable, and the fate of organic matter within the plateau. Lateral DOC fluxes may
represent a significant loss of soil organic carbon, highlighting the importance of
hydrological connectivity in linking terrestrial and aquatic systems. This study emphasizes
the need to account for the relationship between hydrological and chemical processes when
assessing C and GHG fluxes in permafrost-impacted regions.



## 1. Introduction

Northern latitude regions, which store approximately 1,300 Pg of organic carbon (OC) (Hugelius et al., 2014), represent one of the largest terrestrial carbon reservoirs on Earth (Schuur et al., 2008; Schuur et al., 2015; Walter et al., 2006). Sequestered under cold and oxygen-limited conditions, this carbon (C) is increasingly vulnerable to release as permafrost thaws due to climate warming, generating significant feedbacks that complicate predictions of future climate trajectories (Schuur et al., 2008; Schuur et al., 2015; Walter et al., 2006). As permafrost degrades, the release of greenhouse gases, particularly methane ($CH_4$) and carbon dioxide ($CO_2$), through the microbial decomposition of previously frozen organic matter (OM) can rapidly escalate the impact of this feedback (Schuur et al., 2008; Walter et al., 2008; Wik et al., 2016; Zimov et al., 2006). While the large-scale thaw of permafrost is widely recognized (Leppiniemi et al., 2023), the timing, magnitude, and pathways of carbon release remain uncertain, influenced by processes such as burial, mobilization, lateral export, and mineralization (Verdonen et al., 2023; Vonk et al., 2015).

Permafrost thaw leads to irreversible landscape transformations. Peatlands in northern Norway are predominantly located in the sporadic permafrost zone, where they form distinctive landscape features such as peat plateaus and palsas. These are peat uplands and mounds with a frozen core, elevated above the water table by the formation of segregation ice (Alewell et al., 2011; Krüger et al., 2017). As these features degrade, permafrost thaw is often abrupt and subsidence and collapse is to be expected, leading to the formation thermokarst ponds, as excess ground ice is lost (Martin et al., 2021). More than half of the permafrost areas in the Scandinavian Peninsula are at risk of disappearing under current and projected climate conditions (Gisnås et al., 2017; Schuur et al., 2008). The areal extent of peat plateaus in this region decreased by 33%–71% between the 1950s and the 2010s, with rapid degradation observed during the last decade (Borge et al., 2017). This regional degradation





mirrors processes observed across the northern hemisphere, including in the Canadian Arctic,
European Russia, and the Kola Peninsula, highlighting the vulnerability of sporadic
permafrost regions to warming climates (Krutskikh et al., 2023; Payette et al., 2004; Sannel
and Kuhry, 2011). While the processes driving these transformations, such as thermal
disturbances, vegetation shifts, and subsidence, are relatively well-studied, their
consequences for GHG fluxes and C cycling remain uncertain, limiting our ability to project
future climate feedbacks (Holmes et al., 2022; Olefeldt et al., 2021; Turetsky et al., 2020).
Among the new landscape forms that emerge from degrading peat plateaus, thermokarst
ponds and wetlands play a critical role in greenhouse gas dynamics. These small aquatic
systems, formed by the thaw and collapse of permafrost, are characterized by high
concentrations of dissolved organic carbon (DOC) and inorganic carbon (DIC) (Abnizova et
al., 2012; Martin et al., 2021; Matveev et al., 2018). Thermokarst ponds, in particular, act as
hotspots for $CH_4$ and $CO_2$ emissions due to their unique biogeochemical conditions,
including hydrological isolation, anoxic sediments, and high organic matter availability (In 't
Zandt et al., 2020; Polishchuk et al., 2018; Vonk et al., 2015; Ward and Cory, 2015). Despite
their small size, these water bodies can contribute significantly to regional C fluxes, with $CH_4$
and $CO_2$ supersaturation levels often surpassing those of larger lakes or surrounding tundra
ecosystems (Abnizova et al., 2012; Kuhn et al., 2018; Shirokova et al., 2012). However, their
contributions are often overlooked in large-scale C assessments, as their small size makes
them difficult to detect using satellite-based methods (Holgerson and Raymond, 2016; Muster
et al., 2017).
As permafrost thaw progresses, the transition of isolated thermokarst ponds to interconnected
wetland systems further alters GHG dynamics. While northern permafrost wetlands currently
act as a C sink, the inclusion of thaw pond emissions into broader wetland carbon budgets
reveals their potential to offset the sink capacity by 39% (Kuhn et al., 2018). Compared to



thermokarst ponds, wetlands have sustained $CH_4$ fluxes over larger areas due to persistent
waterlogging and OM decomposition (Pirk et al., 2024; Swindles et al., 2015; Turetsky et al.,
2020), thus constituting important long term $CH_4$ sources (Bansal et al., 2023). The
transformation from stable permafrost to thermokarst landscapes is accompanied by shifts in
hydrology, OM lability, and microbial activity, which collectively shape $CO_2$ and $CH_4$
production pathways (Holmes et al., 2022; Laurion et al., 2020). Understanding the dynamics
of these evolving systems is critical for assessing the broader impacts of permafrost thaw on
regional C uptake and emissions as well as global C cycles.
Northern Norway's sporadic permafrost zone, with its abundant small thermokarst ponds and
emerging wetlands, provides a valuable opportunity to study these processes. The region's
rapidly degrading peat plateaus host significant C stocks, yet small aquatic systems,
especially those in Fennoscandia, remain underrepresented in Earth system models
(Abnizova et al., 2012; Muster et al., 2019; Muster et al., 2017). Existing studies emphasize
the importance of quantifying $CH_4$ and $CO_2$ fluxes in these environments and their
implications for C budgets (Abnizova et al., 2012; Matveev et al., 2018). However, critical
questions remain regarding how transitions between permafrost, thermokarst, and wetland
systems influence C dynamics, and whether these landscapes function as net C sources or
sinks under changing climatic conditions (Sim et al., 2021).
This study aims to address these gaps by examining the GHG dynamics and C
biogeochemistry of thermokarst ponds and wetland streams in the sporadic permafrost zone
of northern Norway. Over two years, we measured dissolved $CO_2$ and $CH_4$ concentrations,
water chemistry, and OM lability to evaluate the processes driving C fluxes in these systems.
We hypothesize that (1) thermokarst ponds serve as hotspots of $CH_4$ and $CO_2$ production
relative to wetland streams, (2) the transition from isolated ponds to wetlands significantly
alters GHG emission pathways, driven by shifts in hydrology and OC availability, and (3)



recently mobilized OM presents a labile source of C promoting $CO_2$ production in
thermokarst water bodies compared to wetland streams. By exploring these dynamics, this
study provides insights into the role of small water bodies in permafrost C feedbacks,
advancing our understanding of sub-Arctic and boreal C cycling.




## 2. Methods

2.1 Study area

The Iškoras field site (69.34°N, 25.29°E; 381 m a.s.l.) is a permafrost peatland plateau located in the interior of the Finnmark province, northern Norway, on the Finnmarksvidda plateau (Fig. 1). The region of Finnmarksvidda lies between 300 and 500 m a.s.l. and is characterized by a subarctic continental climate. The topography was shaped by Pleistocene glaciations, which deposited ground moraines, glaciofluvial, and glaciolacustrine sediments (Sollid et al., 1973). The depressions in the landscape are commonly filled with peatlands (Borge et al., 2017), and peat plateaus underlain by permafrost are common.

The Iškoras peat plateau covers an area of approximately 4 ha and is part of a 3.3 km$^2$ subarctic headwater catchment that drains into the Báhkiljohka river (91 km$^2$). Mean annual air temperature and precipitation for the 30-year normal (1991-2020) period was -1.9°C ± 1.0°C, and 513 ± 90 mm, respectively (Table 1). For our study period 2021 to 2022, MAAT and MAP were -1.1°C ± 0.4°C, and 589.5 ± 62.5 (SeNorge, 2023). Iškoras lies within the zone of sporadic permafrost and the peat soils extend down to about 1.5 m in the plateau areas (Kjellman et al., 2018) and active layers depths up to 90 cm. The plateau exhibits a complex surface of intact and degrading palsas, along with thermokarst ponds, and is surrounded by wetlands and a stream to the northwest (Martin et al., 2019). Between 2019 and 2022, up to 0.8 m of subsidence of palsas was measured at localized sites (Pirk et al., 2024). The site is located about 90 km south of the nearest coastal fjord and is dominated by mountain birch forest (*Betula pubescens*) and tundra vegetation, including dwarf birch (*B. nana*). The plateau consists primarily of low heath shrubs, Ericaceae (*Empetrum nigrum, Rhododendron tomentosum*), lichen crusts, mosses, and cloudberry (*Rubus chamaemorus*) or bare ground, while the surrounding wetlands are dominated by Sphagnum mosses, sedges (*Carex* spp.), and cotton grass (*Eriophorum* spp.) (Kjellman et al., 2018; Martin et al., 2019).



| | Unit | Mean ± std for 1991-2020 | Mean + std for 2021-2022 |
|---|---|---|---|
| annual temperature | °C | -1.9 ± 1.0 | -1.1 ± 0.4 |
| summer temperature | °C | 10.4 ± 2.2 | 11.8 ± 0.2 |
| annual precipitation | mm | 513 ± 90 | 589.5 ± 62.5 |
| summer precipitation | mm | 196 ± 53 | 207 ± 48 |

**Table 1 Mean and standard deviation (std) of climate parameters for the Iškoras catchment for the**
**normal period (1991–2020) and the study period 2021-2022.** Summer is defined as May to
September.

The study area included water bodies within a peat plateau and the adjacent wetland, selected
for sampling and monitoring. Measurements and samples were taken approximately monthly
in the ice-free season from May until October in 2021 and 2022. The waterbodies consisted
of three thermokarst ponds (TK-Pond 1, 2, and 3), a seasonal drainage channel (TK-Drain)
connecting the peat plateau to the wetland, and the wetland's inlet and outlet streams (Inlet,
Outlet), with the outlet also marking the terminus of the Iškoras catchment. From mid-May to
early November, monitoring showed that the thermokarst ponds and Inlet were ice-free for
170 ± 5 days, while the Outlet remained ice-free for 184 days (Table SI 1).
The thermokarst ponds varied in hydrological connectivity and permafrost influence,
reflecting differences in age and physical characteristics. TK-Pond 1 (0.4 m depth) is small
and located at the peat plateau–wetland transition, experiencing periodic hydrological
isolation. TK-Pond 2, the largest and deepest (1.5 m depth), lies centrally on the peat plateau,
surrounded by degrading palsas. TK-Pond 3 (0.6 m depth) is situated at the plateau's edge and
was initially isolated by a surrounding permafrost mound.
The TK-Drain is a shallow, ephemeral drainage channel that provides the primary
hydrological connection between the peat plateau and wetland. The wetland's inlet stream
(0.6 m wide, 15–40 cm deep) begins approximately 200 m upstream of the plateau, flowing
through birch forest and mires without permafrost before entering the wetland where the



channel becomes less defined. The outlet stream (0.8 m wide, 30–60 cm deep) re-emerges
approximately 700 m downstream at the wetland's far end, serving as the catchment outlet.
Between September 2020 and October 2022, a total of nine field campaigns were conducted
for regular sampling of water chemistry, dissolved gases, $CO_2$ emissions, dark incubations
and high-frequency monitoring of water height and temperature.

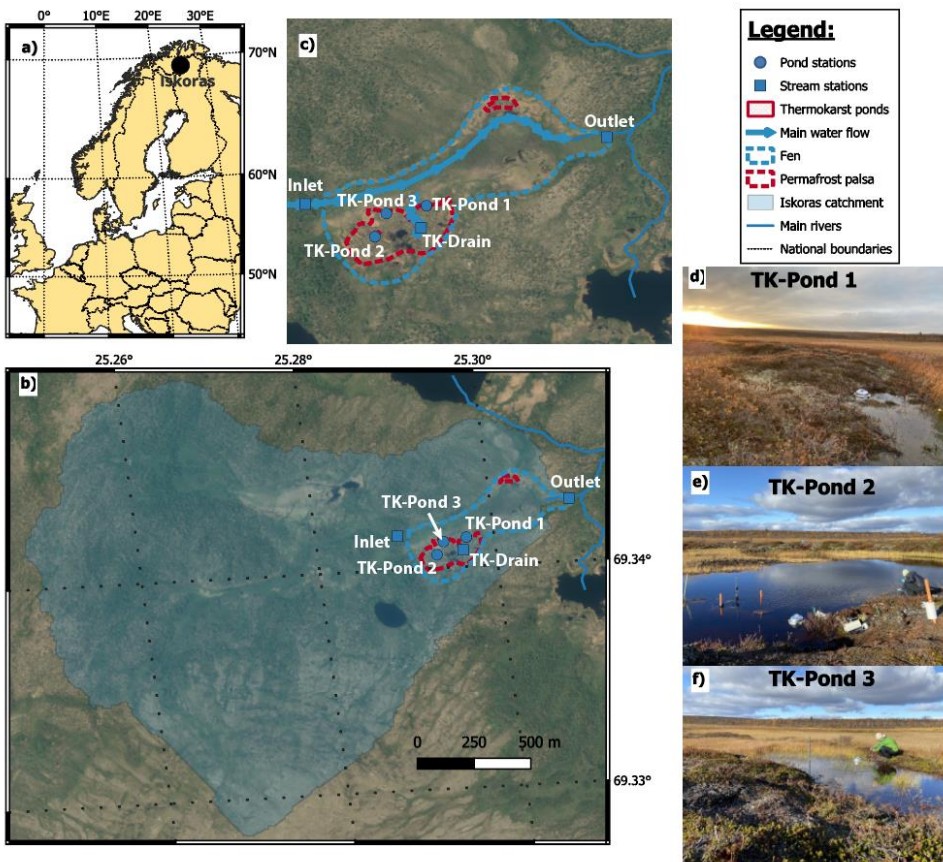


**Figure 1 Location map** of the study area in Europe (a) and the Iškoras catchment (b; determined
from the outlet station Outlet). Close-up of the wetland (c) with regular sampling sites and main
water flow direction. Pictures of the three main pond sites are also shown (d–f).

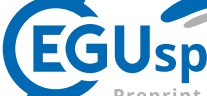



2.2 Water chemistry
Water samples for chemical analyses were collected using standardized procedures. From
each site, 500 mL unfiltered water was collected in HDPE rectangular bottles (Emballator
Melledrud AB, Stockholm, Sweden) after rinsing with sample waters three times, kept dark
after sampling, carried out of the field, and stored within hours after sampling at 4ºC. The
samples were then transported by car and plane back to the laboratory and delivered for
chemical analysis where they were kept at 4ºC until analysis. Chemical analysis of pH,
electrical conductivity (EC), alkalinity, sulfate ($SO_4$), silica ($SiO_2$), ammonium ($NH_4$), nitrate
($NO_3$), total phosphorous (totP), total organic carbon (TOC), DOC (filtered by 0.45 μm) and
particulate organic carbon (POC) (filtered and then combusted at 1800 ℃) in the water
samples was performed at accredited laboratories at the Norwegian Institute for Water
Research (NIVA); methods for analysis and quality control are described in Vogt and
Skancke (2022). The samples were not fully digested according to standard procedures
required for the determination of total nitrogen (totN), hence totN values are expected to be
underestimated and are therefore not shown in the manuscript (Thrane et al., 2020).
Absorption spectra of DOM were measured at NIVA for wavelengths between 200 and 900
nm, using 1 nm intervals, with a 5 cm cuvette length and Milli-Q water as a reference, using a
Lambda 40 UV/Vis spectrophotometer (Perkin Elmer, USA) and expressed in absorbance pr
cm. In two samples, incomplete filtration caused excess scattering, and these spectra were
removed. The absorbencies were used to calculate specific UV absorbency (sUVa =
$A_{\lambda 254nm}$/mg C $L^{-1}$) and the specific UV absorption ratio (SAR = $A_{\lambda 254nm}/A_{\lambda 400nm}$).

2.3 Dissolved gas analysis
Dissolved gases ($CO_2$, $CH_4$) were sampled in the field using the acidified headspace
technique (Åberg and Wallin, 2014). Duplicate gas samples were collected according to



Valiente et al. (2022). Two 50 mL syringes were filled and sealed underwater without air
bubbles to prevent gas loss. Excess water was expelled to retain 30 mL, and 20 mL of
ambient air was drawn in to create a headspace. Samples were acidified with 0.6 mL of 3%
HCl to achieve pH <2, ensuring DIC was present as $CO_2$. Equilibrium was reached by
shaking for one minute, followed by a 30-second rest, repeated thrice. Fifteen mL of
headspace gas was transferred to 12 mL evacuated vials, and water temperature was
measured immediately after. Samples were stored at room temperature and flown to southern
Norway for analysis. A 15 mL ambient air sample was taken daily for background correction.
Analysis was performed via automated gas chromatography (GC) at the Norwegian
University of Life Sciences (NMBU), as described by Yang et al. (2015). A GC autosampler
(GC-Pal, CTC, Switzerland) injected 2 mL headspace samples into an Agilent 7890A GC
(Santa Clara, CA, USA) with a 20-m wide-bore Poraplot Q column at 38°C, using He as the
carrier gas to separate $CH_4$ and $CO_2$ from Ar, $N_2$, and $O_2$. For calibration, certified standards
of $CO_2$ and $CH_4$ in He were used (AGA, Germany) and $N_2$, $O_2$, and Ar were calibrated using
laboratory air. $CH_4$ was measured with a flame ionization detector (FID). A thermal
conductivity detector (TCD) was used to measure all other gases.
Dissolved gas concentrations were calculated from headspace concentrations corrected for
background air, applying temperature-adjusted Henry's law constants (Wilhelm et al., 1977)
based on the recorded water temperature. At pH >4, a non-negligible amount of DIC is in the
form of (bi)carbonates ($HCO_3^-$, $CO_3^{2-}$). The bicarbonate concentrations were calculated based
on pH, dissolved $CO_2$ and the temperature-adjusted first dissociation constant ($pK_1$ = 6.41 at
25°C; Stumm and Morgan (2013)) of the carbonic acid equilibrium. Dissolved $CO_2$ was
calculated as DIC minus bicarbonate. To facilitate comparisons with existing studies that
report dissolved gases in µatm, we converted dissolved gas concentrations to $CO_2$ or $CH_4$





saturation indexes ($GHG_{SI}$) assuming atmospheric partial pressures of $CO_2$ and $CH_4$ as 400
µatm and 1.9 µatm, respectively:
$$GHG_{SI} = \frac{[GHG]}{[GHG]_{saturation}}$$
Where $[GHG]$ is the measured dissolved $CO_2$ or $CH_4$ concentration, and $[GHG]_{saturation}$ is
the concentration of dissolved $CO_2$ or $CH_4$ at equilibrium with their respective atmospheric
partial pressure.

2.4 Diffusive $CO_2$ fluxes from water to atmosphere
Measurements of $CO_2$ fluxes from water to atmosphere (diffusive $CO_2$ fluxes) were measured
at each site for 30-60 minutes using self-made, opaque flux chambers as described by
Bastviken et al. (2015) at the water-air interface. The chamber consists of a Senseair K30
sensor (Senseair AB, Delsbo, Sweden) housed within a plastic bucket that records $pCO_2$,
temperature, and relative humidity every 30 seconds. Fluxes are calculated from the linear
increase in $pCO_2$ corrected for ambient temperature and humidity in the chamber (Bastviken
et al., 2015) considering the internal air volume and the water surface area covered by the
chamber. Single measurements with a linear increase in $pCO_2$ with time associated with a
coefficient of determination ($R^2$) lower than 0.9 were discarded.

2.5 Dark incubations
Water samples were collected for short term dark incubations started directly in the field
lasting between 18 and 30 hours to estimate DOM mineralization and GHG processing rates.
Serum flasks (120 mL) were filled with 80 mL of water with a 50 mL syringe equipped with
a long tube. The syringe was filled and closed under water, and the water was gently pushed
at the bottom of the serum flask to prevent gas loss. The remaining 40 mL were left with
ambient air as headspace. The flasks were crimp-sealed with gas-tight, butyl-rubber septa,



sealed, covered with aluminium foil and kept at field temperature (for maximum 6 hours),
transported back from the field to be stored at room temperature (18-20°C). The day
following the sampling (18 to 30 hours after sampling), the incubations were stopped by
adding 1.6 mL 3% HCl to reach a final pH below 2, after which gas samples were taken
following the protocols described above. Results from the dark incubation were expressed as
rates of DIC production over the course of the incubation period by comparison with initial
DIC concentrations and reported as $\mu M h^{-1}$:
$DIC_{rate} = \frac{[DIC]_f - [DIC]_0}{h}$          (Eq. 2)
where $[DIC]_f$ is the final solute concentrations in the dark incubation and $[DIC]_0$ is the initial
solute concentration taken in the field (see Sect 2.3) in $\mu M$, and $h$ is the incubation duration
in hours. In addition, we normalized the DIC production rate with DOC concentration to
estimate DOM mineralization rates (per time unit). Also, we calculated the first-order DOM
decay rate ($yr^{-1}$) using the exponential decay rate model (Mostovaya et al., 2017). The
exponential decay model, based on early studies on sediment diagenesis (Boudreau and
Ruddick, 1991; Westrich and Berner, 1984), is often the best model to describe decay rates
from bioassays in closed systems (Vähätalo et al., 2010) and has been widely used to describe
DOM degradation reactions. Under the exponential decay model, the decay constant ($k_{DOM}$;
$yr^{-1}$) can be expressed as:
$k_{DOM} = ln\left(\frac{DOC}{DOC - ([DIC]_f - [DIC]_0)M_C}\right) \times \frac{8766}{h}$          (Eq. 3)
where $DOC$ is the DOC concentration in $\mu g\ L^{-1}$, $M_C$ is the molecular mass of C in $g\ mol^{-1}$ and
8766 is the number of hours in a year. Where $[DIC]_f$ was equal to or below $[DIC]_0$, we
removed the values from the dataset assuming that the temperature correction of $[DIC]_0$ was
not precise enough (three of 39 samples) to allow quantification of $CO_2$ processing rates.



These occurred in September 2020 and October 2021, under cold field conditions, when
$[DIC]_0$ was overestimated because of unknown sample temperature in the field.

2.6 Statistical methods
Statistical analyses were conducted to evaluate differences between sites for various
measured parameters. One-way analysis of variance (ANOVA) was employed to test for
differences among groups. Pairwise comparisons of group means were performed using
Student's t-test using JMP 18.0.11 (2024 JMP Statistical Discovery LLC). For data that did
not conform to normal distribution assumptions, non-parametric methods were applied,
specifically the Wilcoxon rank-sum test, to ensure robust comparisons across sites. Results
are displayed in the form of connecting letters reports within the tables. Sites with the same
letter (e.g., "A" or "B") indicate no statistically significant differences in the measured
parameter between those groups at the $p < 0.05$ significance level. Groups with different
letters (e.g., "A" vs. "B") are significantly different. When overlapping letters (e.g., "AB") are
reported, those groups are statistically similar to others with at least one shared letter but may
differ from groups with entirely distinct letters. Figures were created using the ggplot2
package (Wickham, 2016) using R software (R Core Team, 2021).



## 3. Results

3.1 Water chemistry

The thermokarst water bodies were more acidic, richer in DOC and total P, and lower in $SO_4$ and $SiO_2$ compared with the wetland streams (all differences statistically significant; Table 2; Fig. 2). The low pH of the ponds is consistent with their high DOC, and thus high organic acidity. The water bodies aligned along the inverse DOC-pH relationship with TK-Pond 3 at the top, followed by TK-Pond 2 and TK-Pond 1. The TK-Drain usually held an intermediate position between the thermokarst ponds and the wetland streams, which were found at the high pH – low DOC end of the curve. Similar patterns were found for DOC-$SO_4$ and DOC-$SiO_2$ relationships (Fig. 2). Particulate OC was on average <5% of TOC in the wetland streams, while POC showed considerably more variation in the thermokarst water bodies, possibly related to inputs from destabilized organic matter from the thawing permafrost.

All water bodies had $NO_3$ concentrations at, or close to, the detection limit, while the thermokarst water bodies had considerable levels of $NH_4$ contrary to the wetland streams (Table 2). Despite incomplete digestion prior to totN determination, totN values were enough to confirm that the dominant form of N was organic. Total P was highest, and most variable, in the ponds which to some extent mirrored the pattern in DOC, understandably given that in these nutrient-poor sites most P would be in an organic form just like N.

The DOM quality indicator SAR was highest in the thermokarst ponds (p<0.03). SAR was positively strongly correlated with DOC concentration (positive, $R^2$ 0.57, p<0.0001), implying that lowest SAR was found in the wetland streams. Other DOM quality indicators (sUVa, associated with aromaticity) tended to be somewhat higher in the wetland streams but did not show significant differences between wetlands and thermokarst water bodies.



**Table 2. Water chemistry parameters for thermokarst ponds and wetland sites during nine**
**sampling campaign.** Median values with standard deviations are shown for all water chemistry
variables, except for pH, which is shown as the median with minimum and maximum values. EC:
electrical conductivity; SO$_4$: Sulfate; SiO$_2$: silica; DOC: dissolved organic carbon; sUVa: specific UV
absorbency, SAR: specific UV absorption ratio; TOC: total organic carbon; NH$_4$: ammonium; NO$_3$:
nitrate, totP: total organic phosphorous; POC: particulate organic carbon (POC, % of TOC). Letters
indicate significant differences between sites for each variable (Tukey's *t*-test, pairwise comparisons,
*p*<0.05; see Sect. 2.6).

| | pH | | EC mS m$^{-1}$ | | SO$_4$ mg SO$_4$ L$^{-1}$ | | SiO$_2$ mg SiO$_2$ L$^{-1}$ | |
|---|---|---|---|---|---|---|---|---|
| TK-Pond 1 | 4.49 (4.16-4.79) | B | 2.1 (0.7) | B | 0.12 (0.07) | C | 2.4 (1.8) | BC |
| TK-Pond 2 | 4.23 (4.03-4.37) | C | 3.2 (0.7) | A | 0.18 (0.20) | C | 1.0 (1.0) | C |
| TK-Pond 3 | 4.06 (3.79-4.32) | C | 4.0 (1.6) | A | 0.11 (0.02) | C | 4.3 (1.7) | B |
| TK-Drain | 4.79 (4.63-4.88) | B | 1.6 (0.1) | B | 0.16 (0.08) | C | 2.3 (1.9) | BC |
| Inlet | 6.69 (6.11-7.36) | A | 2.0 (0.5) | B | 0.85 (0.18) | A | 8.7 (2.4) | A |
| Outlet | 6.56 (6.04-6.97) | A | 1.9 (0.3) | B | 0.60 (0.25) | B | 8.6 (3.0) | A |

| | DOC mg C L$^{-1}$ | | sUVa A$_{\lambda254nm}$/mg C L$^{-1}$ | | SAR A$_{\lambda254nm}$/A$_{\lambda400nm}$ | | TOC mg C L$^{-1}$ | |
|---|---|---|---|---|---|---|---|---|
| TK-Pond 1 | 19.2 (4.4) | C | 3.9 (0.4) | ABC | 8.4 (0.3) | BC | 20.8 (5.1) | C |
| TK-Pond 2 | 25.7 (6.4) | B | 4.2 (0.3) | A | 8.8 (0.5) | AB | 27.4 (7.7) | B |
| TK-Pond 3 | 34.1 (6.8) | A | 3.6 (0.6) | BC | 9.2 (0.7) | A | 34.8 (9.4) | A |
| TK-Drain | 17.3 (6.6) | C | 3.8 (0.4) | C | 8.1 (0.2) | AB | 19.5 (4.5) | C |
| Inlet | 8.5 (2.1) | D | 4.1 (0.4) | AB | 7.6 (0.2) | C | 8.4 (1.6) | D |
| Outlet | 9.0 (1.5) | D | 4.3 (0.3) | A | 7.8 (0.2) | C | 9.4 (1.6) | D |

| | NH$_4$ µg N L$^{-1}$ | | NO$_4$ µg N L$^{-1}$ | | TotP µg P L$^{-1}$ | | POC % of TOC | |
|---|---|---|---|---|---|---|---|---|
| TK-Pond 1 | 23 (26) | B | 2.0 (0.0) | A | 27 (17) | B | 5.6 (3.0) | B |
| TK-Pond 2 | 94 (105) | A | 2.1 (0.3) | A | 18 (9) | BC | 6.1 (4.1) | B |
| TK-Pond 3 | 38 (77) | AB | 1.9 (0.3) | A | 54 (32) | A | 8.2 (5.9) | B |
| TK-Drain | 33 (20) | B | 2.0 (0.0) | A | 20 (18) | BC | 17.1 (11.6) | A |
| Inlet | 2 (2) | B | 1.9 (0.3) | A | 9 (6) | C | 4.4 (1.8) | B |
| Outlet | 4 (8) | B | 1.9 (0.3) | A | 7 (4) | C | 4.1 (1.9) | B |




The inverse relationship between DOC and pH points towards organic acidity as a strong
driver of pH. Additionally, the near-to-neutral pH in the wetland streams is consistent with
groundwater influences from the catchment, which also would explain the elevated $SiO_2$ and
$SO_4$ concentrations. A limited set of water samples were analysed for base cations (Table SI
3), confirming that these were highest in the wetland streams.
The water chemical composition of the ponds mirrored the impact of thawing permafrost: the
TK-Pond 3 is hydrologically most isolated with the lowest pH, highest conductivity, and
highest DOC. TK-Pond 1, located at the transition from peat plateau to wetland, had a higher
pH and lower EC, DOC and $NH_4$ than the other ponds, which is consistent with some
hydrological influences from the wetland and hence less permafrost impact. TK-Pond 2 is
located in the middle of the peat plateau and is by far the largest pond and, under wet
conditions, hydrologically connected to neighbouring ponds. The water chemistry of TK-
Drain was usually most similar to that of TK-Pond 1. An example of pH and EC gradients
from the peat plateau into the wetland is consistent with the influence of thermokarst
waterbodies gradually becoming less dominant in the transition from the peat plateau
complex to the wetland (Fig. SI1).



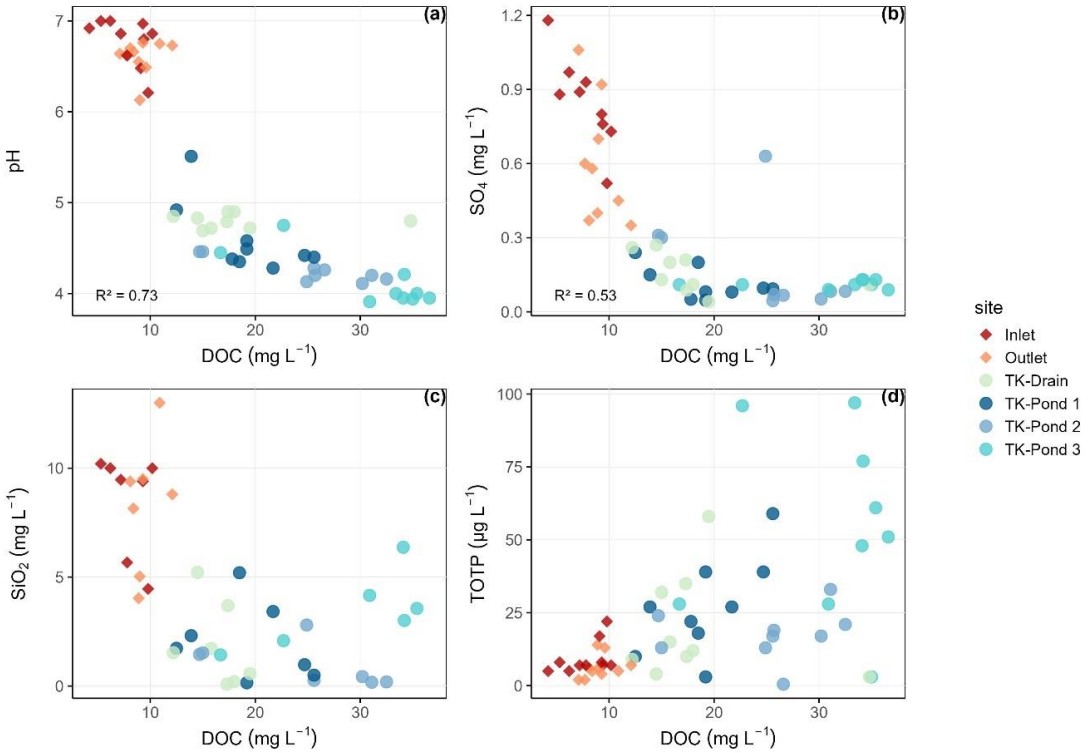


**Figure 2. Relationships between Dissolved Organic Carbon (DOC) and various water quality parameters across different sites.** The scatter plots demonstrate the relationships between dissolved organic carbon (DOC) and pH (a), sulfate ($SO_4$) (b), silica ($SiO_2$) (c), and total organic phosphorus (totP) (d).






3.2 Dissolved gases and gas evasion
All water bodies were oxygenated and dissolved $O_2$ concentrations were on average 61 to
81% of water $O_2$ saturation (Table 3). The ponds are shallow which allow for wind mixing
and they host sphagnum, suggesting active $O_2$ production through photosynthesis. All water
bodies were oversaturated with $CH_4$ and $CO_2$. Dissolved $CH_4$ concentrations were 2000–5000
and ~30 times higher than atmospheric equilibrium, in the ponds and in the wetland streams,



respectively, indicating that all water bodies – thermokarst ponds in particular - are net
sources of $CH_4$ to the atmosphere. The lower $CH_4$ oversaturation in streams compared with
ponds is likely related to higher $CH_4$ losses caused by stream turbulence and/or higher
production rates of $CH_4$ in the thermokarst ponds (Fig. 3).

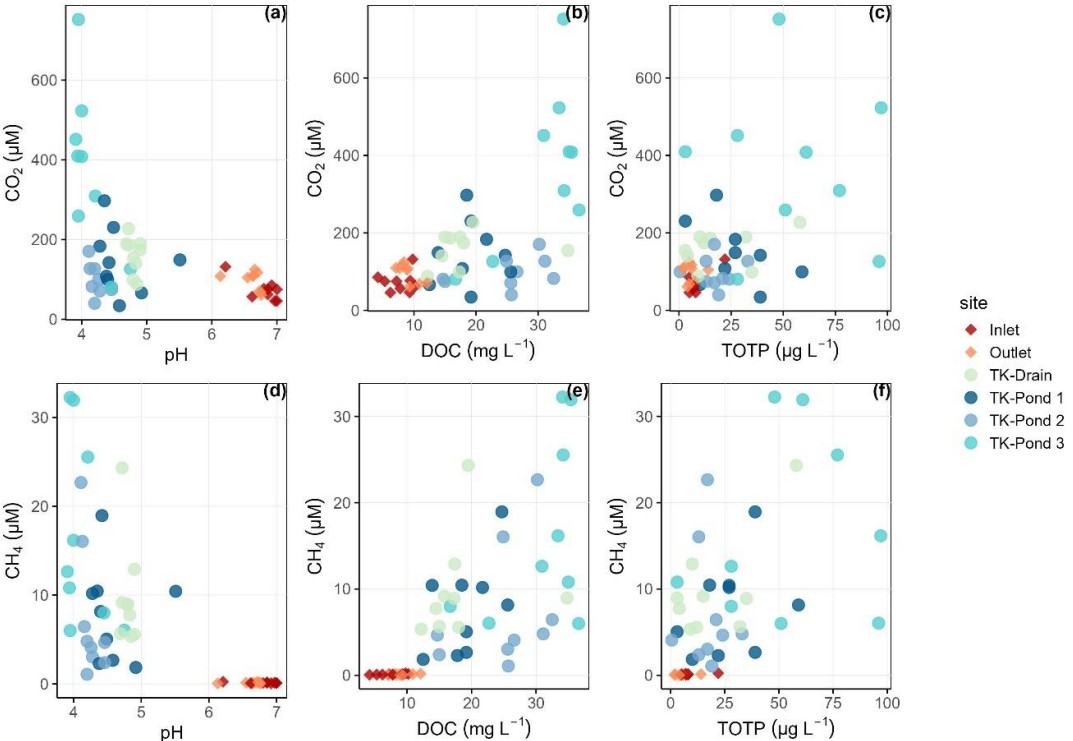


**Figure 3 Variations in carbon dioxide ($CO_2$) and methane ($CH_4$) concentrations in relation to pH,**
**DOC, and totP across sampling sites.** The upper panels show the relationships between $CO_2$
concentrations (µM) and pH (a), dissolved organic carbon (DOC, mg L$^{-1}$) (b), and total organic
phosphorous (totP, µg L$^{-1}$) (c).
$CO_2$ saturation indexes in the thermokarst waterbodies reached 5 to 20 while in the streams
they ranged between 3 and 4. By contrast, the $CO_2$ evasion from the streams (e.g. 1-2 g C m$^{-2}$
day$^{-1}$) was higher than from the ponds (0.3-0.5 C m$^{-2}$ day$^{-1}$), consistent with the higher
turbulence in the streams, and the replenishment of $CO_2$ from bicarbonates from groundwater

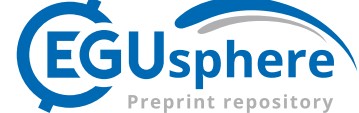

in these streams, which is a geological rather than a recent source of $CO_2$. Bicarbonates
contributed about 60-70% to DIC in the wetland streams (with pH between 6.0 and 7.4),
while bicarbonates in thermokarst water bodies were almost negligible (with pH below 4.5),
which is consistent with equilibrium between bicarbonates and $CO_2$ over these pH ranges.
TK-pond 3 had the lowest $O_2$ concentrations and the highest $CH_4$ and $CO_2$ concentrations of
all thermokarst water bodies. Concentrations of $CH_4$ and DIC were positively related in the
thermokarst water bodies (r2 0.48, p<0.0001, F-test) but not in the wetland streams (Fig. 4).

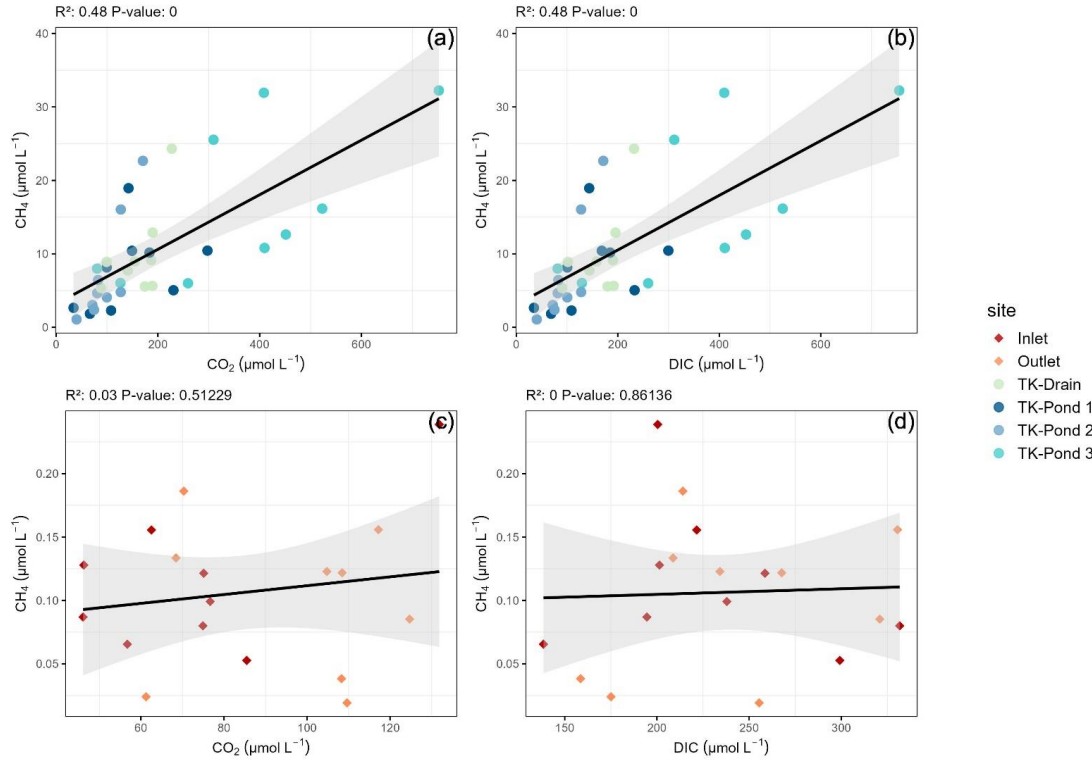


**Figure 4** Relationships between $CH_4$ and (a) $CO_2$, (b) DIC for thermokarst waterbodies, and (c) $CO_2$,
(d) DIC for Inlet and Outlet sites including linear regression lines and corresponding $R^2$ and p-value
statistics. Note scale differences for $CH_4$ between thermokarst waterbodies and the wetland
streams.





The $CO_2$ emissions from the stream sites (Inlet and Outlet) were substantially larger than
those from the thermokarst waterbodies (Table 3). The mean $CO_2$ emission at the Inlet site
was $1.12 \pm 0.46$ g C m$^{-2}$ day$^{-1}$, and at the Outlet site $2.20 \pm 1.15$ g C m$^{-2}$ day$^{-1}$. These values
are 3 to 7 times higher than the fluxes observed from the thermokarst waterbodies, which
ranged from $0.30 \pm 0.22$ g C m$^{-2}$ day$^{-1}$ (TK-Pond 2) to $0.51 \pm 0.28$ g C m$^{-2}$ day$^{-1}$ (TK-Pond 3).
Annual $CO_2$ fluxes for the ice-free period, assuming negligible flux during the ice-covered
months, ranged between 51 g C m$^{-2}$ yr$^{-1}$ and 87 g C m$^{-2}$ yr$^{-1}$ for the thermokarst ponds, and
the streams ranged between 190 g C m$^{-2}$ yr$^{-1}$ (Inlet) and 405 g C m$^{-2}$ yr$^{-1}$ (Outlet).
**Table 3. Mean and standard deviations of dissolved gas concentrations and associated metrics for**
**thermokarst ponds and wetland sites across nine sampling campaigns.** Mean concentrations of $CO_2$
(µM) and $CH_4$ (µM) with their respective saturation ratios, along with $CO_2$ emission flux (g C m$^{-2}$ day$^{-1}$)
), DIC (µM), and oxygen concentrations (µM) with percent saturation. The saturation ratio is defined
as the concentration divided by the equilibrium concentration between the atmosphere and water
at the given temperature. For this study, DIC is considered the sum of dissolved $CO_2$ and
bicarbonate. Letters indicate significant differences between sites for each variable.

| | $CO_2$ | | DIC | $CO_2$ emission |
|---|---|---|---|---|
| | µmol L$^{-1}$ | Saturation ratio $CO_2$ | µmol L$^{-1}$ | g C m$^{-2}$ day$^{-1}$ |
| TK-Pond 1 | 146 (82) B | 7.0 (3.2) BC | 149 (83) BC | 0.36 (0.28) C |
| TK-Pond 2 | 97 (39) B | 5.1 (2.2) BC | 98 (39) C | 0.30 (0.22) C |
| TK-Pond 3 | 369 (206) A | 18.8 (11.3) A | 371 (206) A | 0.51 (0.28) C |
| TK-Drain | 161 (45) B | 8.1 (2.7) B | 165 (46) BC | 0.37 (0.15) C |
| Inlet | 73 (26) B | 3.1 (1.0) C | 232 (59) B | 1.12 (0.46) B |
| Outlet | 97 (24) B | 4.4 (1.1) BC | 241 (59) B | 2.20 (1.15) A |

| | $CH_4$ | | $O_2$ | |
|---|---|---|---|---|
| | µmol L$^{-1}$ | $CH_4$ saturation ratio | µmol L$^{-1}$ | % saturation |
| TK-Pond 1 | 7.8 (5.5) B | 2 330 (1 719) B | 273 (59) AB | 81 (14) A |
| TK-Pond 2 | 7.2 (7.2) B | 2150 (2 110) B | 266 (85) AB | 79 (19) A |
| TK-Pond 3 | 16.6 (10.6) A | 5 109 (3 470) A | 210 (89) B | 61 (22) B |
| TK-Drain | 9.8 (5.9) B | 2976 (1973) B | 260 (65) AB | 77 (18) AB |
| Inlet | 0.1 (0.1) C | 30 (15) C | 297 (59) A | 81 (14) A |



| | | | | |
|---|---|---|---|---|
| Outlet | 0.1 (0.1) C | 28 (18) C | 241 (63) AB | 67 (14) AB |


3.3 DOM processing rates
Average DIC production rates in the different water bodies were highly variable (5.8 – 35.7
$\mu$M day$^{-1}$, Table 4), but tended to be highest in the thermokarst ponds compared with the
wetland streams, while the TK-drain had the lowest rates (Tukey's t-test, p<0.05. The non-
parametric Wilcoxon tests supported these trends, confirming minimal site-specific effects
overall, with TK-Drain showing lower activity). These results reflect the in-situ processing of
DOM in both thermokarst ponds and streams. The DOM mineralization rate did not vary
significantly between sites and neither did the exponential decay rate kDOM (Table 4).
kDOM in the thermokarst ponds ranged from 5.5 yr$^{-1}$ to 6.7 yr$^{-1}$ (Table 4), while the stream
showed higher kDOM values from 8.3 yr$^{-1}$ to 8.8 yr$^{-1}$. The TK-Drain site was substantially
lower (2.0 yr$^{-1}$).
**Table 4. Rates of production and decay**. DIC rates reflect DIC production. The DIC rate/DOC ratio
indicates the relative efficiency of converting DOC to DIC, while kDOM indicates the exponential
decay rate of DOM, showing how quickly DOM is decomposed over time. Letters indicate significant
differences between sites for each variable (*p*<0.05).

| | DIC rate $\mu$mol day$^{-1}$ | | DIC rate/DOC $\mu$mol g C$^{-1}$ day$^{-1}$ | | kDOM yr$^{-1}$ | |
|---|---|---|---|---|---|---|
| TK-Pond 1 | 26.7 (13.7) | AB | 62.9 (34.2) | A | 6.7 (3.7) | A |
| TK-Pond 2 | 34.5 (39.5) | B | 62.3 (74.3) | A | 6.7 (8.2) | A |
| TK-Pond 3 | 35.7 (33.9) | A | 51.2 (36.3) | A | 5.5 (4.0) | A |
| TK-Drain | 5.8 (6.1) | B | 19.2 (21.2) | A | 2.0 (2.3) | A |
| Inlet | 18.8 (11.3) | B | 77.3 (27.5) | A | 8.3 (3.0) | A |
| Outlet | 20.5 (16.7) | AB | 82.0 (64.7) | A | 8.8 (7.0) | A |

4. Discussion
4.1 Thermokarst ponds as hotspots of methane emissions
Thermokarst ponds in Iškoras display CH$_4$ saturation indexes of 2300 to 5000, which is
among the highest values reported in the literature for natural waterbodies. These findings





align with Shirokova et al. (2012) and Matveev et al. (2018) who documented saturation
indexes of 50 to 5000 in Siberian thermokarst depressions, and of 5 to 50 in subarctic lithalsa
lakes, respectively. Such high $CH_4$ concentrations in these poorly connected, small, and
relatively protected water bodies are consistent with the established inverse relationship
between $CH_4$ concentrations and water body size, hydrological connectivity, and turbulence
exposure (Abnizova et al., 2012; Kankaala et al., 2013; Polishchuk et al., 2018). Similarly, at
Iškoras, the smallest pond exhibited the highest $CH_4$ oversaturation, in combination with the
highest DOC, totP, and lowest pH values, likely linked to destabilization of thawing
permafrost combined with limited hydrological connectivity. In fact, low pH and elevated
totP concentrations are commonly related to increased DOM concentrations (Holmes et al.,
2022; Ward and Cory, 2015), which could originate from destabilized permafrost (Turetsky
et al., 2020).
Thermokarst ponds were highly oversaturated in $CH_4$ despite the presence of dissolved $O_2$.
$CH_4$ production is known to occur mainly in anoxic sediments (Bastviken et al., 2004; Clayer
et al., 2016; Wik et al., 2016), from where $CH_4$ is subsequently transported to overlying
water. The microbial activity responsible for $CH_4$ production may be enhanced by fresh OM
input from ongoing thermokarst development (Crevecoeur et al., 2017), as recently observed
in laboratory incubations with recently inundated peat material from the Iškoras site (Kjær,

2024).

4.2 Impact of thawing peat plateaus on water chemistry
The transition of permafrost-underlain peat plateaus to thermokarst ponds and further to
wetlands can markedly shift the landscape-scale GHG dynamics as permafrost continues to
thaw (Hugelius et al., 2020; Sannel and Kuhry, 2011). Thermokarst ponds are localized $CH_4$
hotspots; however, as these systems transition into wetlands, emissions patterns change due
to altered hydrology and biogeochemistry (Holmes et al., 2022; Peura et al., 2019). In



wetlands, persistent inundation creates anoxic conditions favourable for methanogenesis,
often leading to significant $CH_4$ emissions (Cui et al., 2024). Pirk et al. (2024) demonstrated
that fens at Iškoras, as an example of such inundated wetland systems, emit large amounts of
$CH_4$, particularly where fresh organic carbon is available from peat decomposition from
degrading palsa edges. Similarly, Turetsky et al. (2020) noted that newly formed wetlands,
arising from permafrost thaw become additional $CH_4$ sources due to labile OC availability.
Holmes et al. (2022) observed that inundation in permafrost landscapes increases $CH_4$
emissions because of waterlogged conditions that limit oxygen diffusion and enhance
anaerobic decomposition. Additionally, Kjær et al. (2024) highlighted that recently thawed
permafrost peat in wetland systems, including peat from Iškoras, features a high $CH_4$
production potential due to the presence of labile C and methanogenic microbial
communities.
Unlike thermokarst ponds, which are spatially limited, wetlands have a wider spatial extent.
The carbon balance in wetlands depends not only on $CH_4$ fluxes but also on changes in $CO_2$
dynamics as organic matter mineralization occurs under waterlogged conditions (Turetsky et
al., 2020). In this context, the transition from thermokarst ponds to wetlands represents a shift
from localized emissions of both $CH_4$ and $CO_2$ emissions to more spatially homogenous $CH_4$
emissions, while $CO_2$ is sequestered (Pirk et al., 2024).
Although a focus on vertical fluxes dominates many studies, lateral DOC fluxes also play a
role in carbon dynamics in permafrost-affected systems. Wetland and peatland ecosystems
exhibit high rates of lateral DOC export due to hydrological connectivity (Tank et al., 2018).
Beckebanze et al. (2022) reported that lateral fluxes, though smaller in magnitude compared
to vertical fluxes, are essential in carbon budgets, potentially contributing to carbon loss
especially during peak runoff events. These lateral DOC fluxes transport carbon from soils to



aquatic systems, where it may later contribute to downstream GHG emissions or C
sequestration. The potential for fens and wetlands to transition into significant GHG
contributors and increased hydrological connectivity leading to greater potential lateral C
fluxes highlights the need for integrated C budget models that capture the evolving landscape
dynamics in permafrost regions.
4.3 Carbon dioxide dynamics in thermokarst water bodies
Both thermokarst ponds and streams in Iškoras are oversaturated with $CO_2$, a common feature
of Arctic and subarctic aquatic systems (Allesson et al., 2022; Bastviken et al., 2004).
However, the mechanisms driving $CO_2$ fluxes differ between ponds and streams. In ponds,
despite high $CO_2$ concentrations, $CO_2$ release is lower than in streams and likely limited by
the lack of turbulence. By contrast, streams exhibit enhanced $CO_2$ fluxes due to high
turbulence and carbonate inputs, as described by Zolkos et al. (2019), who emphasized the
role of groundwater-derived bicarbonates in sustaining $CO_2$ fluxes in Arctic streams.
Quantitatively, $CO_2$ efflux from streams at Iškoras averages 0.4 g C m$^{-2}$ day$^{-1}$, aligning
closely with values observed in Siberian permafrost streams (0.3–0.5 g C m$^{-2}$ day$^{-1}$;
(Shirokova et al., 2012)). This flux reflects the significant contribution of turbulent flow and
bicarbonate-rich groundwater inputs, processes shown to facilitate $CO_2$ release (Lundin et al.,
2013; Raymond et al., 2013). Streams benefit from continuous replenishment of $CO_2$ from
bicarbonates, which account for 60–70% of DIC in these environments (Wallin et al., 2018;
Zolkos and Tank, 2020).
In contrast, $CO_2$ emissions from ponds at Iškoras are notably lower, similar to findings of
Campeau and Del Giorgio (2014), attributed to the ponds' high DOC-to-bicarbonate ratios,
which restrict bicarbonate formation and subsequent $CO_2$ production (Abnizova et al., 2012;
Bastviken et al., 2004). The limited water mixing in ponds further diminishes $CO_2$ flux due to
low gas exchange rates compared to streams. However, DIC production rates in ponds (26.7–





35.7 μM day$^{-1}$) remain sufficient to sustain $CO_2$ effluxes of 5–10 mol C m$^{-2}$ year$^{-1}$, consistent
with findings by (Shirokova et al., 2012), who linked DIC production directly to DOC
concentrations in Arctic aquatic systems. These DIC production rates are relatively high
compared to other reported values in the literature (Catalán et al., 2016).
High $CO_2$ evasion from lower-order streams is caused by high flow velocities and associated
turbulence causing high gas exchanges (Schelker et al., 2016). These drivers likely explain
the relatively higher $CO_2$ efflux from Iškoras streams compared to ponds. Furthermore, DOM
lability, influenced by pH and nutrients, plays a critical role. High molecular-weight DOM, as
indicated by low SUVA and SAR values, can inhibit microbial oxidation, thereby slowing
DOM decay rates (Shirokova et al., 2019). Ward and Cory (2015) also noted that DOM from
thawing permafrost, while less aromatic and more labile compared to active layer DOM, may
become limited by environmental factors such as pH and nutrient availability, resulting in
lower mineralization rates. The acidic conditions in ponds could shift microbial communities
and affect activity (Vigneron et al., 2019), a factor that can further limit $CO_2$ fluxes from
ponds in addition to the low gas exchange rate.
In summary, the higher $CO_2$ fluxes observed in streams at Iškoras are likely driven by the
combined effects of turbulent flow, bicarbonate input, and mineral weathering, whereas lower
emissions from ponds are shaped by organic acidity, limited hydrological connectivity, and
lack of turbulence. These dynamics emphasize the need to account for the interplay of
hydrological and chemical factors when assessing C fluxes in permafrost-impacted regions.
4.4 Climate feedback implications
The distinct roles of $CH_4$ and $CO_2$ in the Iškoras landscape underscore their unique climate
feedback potentials. The transition from thermokarst ponds to wetlands modifies the overall
GHG footprint of the peatland-wetland continuum, balancing the loss of localized $CH_4$
emission hotspots with the emergence of sustained, long-term $CH_4$ emissions from wetlands,



while the fate of organic matter currently stored in permafrost remains uncertain. At the same
time, $CO_2$ fluxes from the streams and rivers may increase due to enhanced hydrological
connectivity and increased organic matter input (Zolkos et al., 2019), in agreement with the
results of our study. These findings reflect the complex interplay of ecological and
hydrological factors shaping GHG emissions in permafrost landscapes. Turetsky et al. (2020)
and Pirk et al. (2024) both emphasized the need for further research on spatiotemporal
variability, particularly during thaw cycles, to refine predictions of permafrost C feedbacks.

## 5. Conclusions

This study highlights the distinct biogeochemical roles of thermokarst ponds and wetland
streams in a landscape of sporadic permafrost in subarctic Norway. Thermokarst ponds at the
Iškoras site, characterized by low pH, high organic acidity, and elevated DOC concentrations,
are currently hotspots for $CH_4$ emissions, with stable DOM lability driving sustained carbon
processing. In contrast, wetland streams exhibit higher $CO_2$ fluxes, largely driven by
turbulence and bicarbonate replenishment from groundwater. Despite similarities in DOM
mineralization rates between ponds and streams, environmental constraints, such as pH,
microbial community composition, and hydrodynamic mixing, are likely controls of the
observed differences in GHG fluxes. As thermokarst ponds transition into wetlands, they will
no longer function as hotspots for $CH_4$ emissions. Instead, $CH_4$ emissions are likely to
increase across the entire landscape, as sustained waterlogging promotes elevated $CH_4$
production. These ecological shifts, coupled with lateral DOC losses from peat plateaus,
highlight the importance of hydrological connectivity in linking terrestrial and aquatic C
dynamics. Such transitions emphasize the need for integrated C budget models that account
for the evolving contributions of small aquatic systems to regional and global C cycles.



Future research should prioritize direct measurements of $CH_4$ fluxes, microbial community
contributions to DOM decomposition under varying environmental constraints, and the
temporal variability of gas production and emissions. Additionally, exploring seasonal
dynamics, lateral carbon transport, and hydrological processes will provide critical insights
into C cycling. Investigating catchment-scale signals, such as DOC concentrations across
entire river systems and their links to permafrost contributions, can further advance our
understanding of landscape-level processes. By addressing these questions, we can better
predict the trajectory of permafrost-impacted landscapes and their feedbacks to the global C
cycle in a warming climate.
## Data availability
All data supporting this study will be made available on a permanent repository upon
acceptance.
## Author contributions
JKK and HDW conceptualized the study. JKK, HDW and FC participated in data collection.
JKK, FC, HDW and PD conducted the experiments and performed data analysis. JKK, HDW
and FC created the figures. JKK and HDW drafted the initial manuscript, and HDW, FC, SW
and PD revised and edited the final version.
## Competing interests
The authors declare that they have no conflict of interest.
## Acknowledgements



NIVA core funding (Research Council of Norway, contract nr 342628/L10) Global Change at
Northern Latitudes, BIOGOV project (Research Council of Norway, project nr 323945), Uta
Brandt, Hanna Lee, Inge Althuizen, Emelie Forsman



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
