# Peer review of "Water chemistry and greenhouse gas concentrations in waterbodies of a thawing permafrost"

_EGUsphere, 2025_

## Author Comment (AC2)

**Response to RC2**

Thank you for your thorough review our manuscript, *Water chemistry and greenhouse gas concentrations in waterbodies of a thawing permafrost peatland complex in northern Norway*. We are pleased that you also recognize the value of our dataset, particularly regarding the greenhouse gas (GHG) concentrations in water bodies from this emerging environment in northern Norway. We respectfully disagree with your suggestion to remove the water chemistry data from this manuscript. In our reply, we provide arguments that the water chemistry data in this dataset are of good quality and present valuable insights into the hydrochemical contrasts between the water bodies we studied, which in turn presents useful context for the GHG data and C cycling. In the following response, comments from the reviewer are *italicized in blue text* and our responses and edits are in standard, black text.

Kind regards on behalf of all authors,

Jacqueline Knutson

*The manuscript « Water chemistry and greenhouse gas concentrations in waterbodies of a thawing permafrost peatland complex in northern Norway » presents original data of greenhouse gas (GHG) concentrations in water bodies from a peatland area in the permafrost region in Norway collected over 2 years of monitoring. The dataset for the GHG is highly valuable for the community given the little knowledge on these emerging environments. The data are accompanied with water chemical composition, but some additional information is needed before these additional data can be used in the manuscript (see comments below). I would really encourage the authors to address the comments below to allow the publication of this highly valuable GHG dataset.*

*Major comments*

*The most important comments are about the water chemistry measurements (sampling, analytical procedures, and data availability).*

*Firstly, according to the method description, the unfiltered waters were sampled in HDPE bottles and kept at 4°C until analysis. The method does not specify whether filtration was performed. For the analysis of SiO2, SO4, NO3, NH4, totP concentrations, if filtration was performed, it should be specified with what type of filter (material), what pore size, and how long after sampling. For DOC and POC analysis, the information about filtration should be more specific and specify which type of filter, pore size, and how long after sampling. Importantly, for the analysis of SiO2, SO4, NO3, NH4, totP concentrations, if no filtration was performed, I would strongly recommend not to use the data, and to focus the manuscript on the C cycling. Unfiltered waters may contain small sized particles which continue to interact with the dissolved phase after sampling (dissolution, adsorption, etc). This is why it is conventionally recommended for this type of water analysis to filter within 24h to 48h.*

We would like to respond to the major comment raised regarding the place of the water chemical data in the manuscript. We applied generally accepted methods for sampling and analyzing freshwaters in this manuscript but may not have described these methods in sufficient detail, for which we stand corrected. To assess whether delayed filtration influenced our results, we conducted a filtration comparison experiment in September 2022. This test showed no significant differences between field-filtered and lab-filtered samples for DOC, pH, $SiO_2$, $SO_4$, TOC, TOTN, or conductivity. Minor differences were observed for $PO_4$-P and total P in TK-Pond 1 and TK-Pond 3, likely due to particle flocculation between collection and filtration. A more notable difference was found in $NH_4$-N at TK-Pond 1, suggesting some variability in ammonium concentration. However, overall, the results indicate that our sampling and filtration approach provided reliable data. We found that this justified our sampling methods, e.g. sample water in HDPE bottles, keep them cool (4°C) and dark, transport them as quickly as possible to the lab and do filtration for some parameters in the laboratory. The field

filtration that the reviewer suggests would have taken several hours extra per field day, which would have limited other data collection. Given these findings, we maintain that our water chemistry dataset is scientifically robust and suitable for inclusion in the manuscript. To improve clarity and transparency, we propose to update the methods section with more detailed information on laboratory procedures and have added the filtration comparison experiment results in the supplementary material. Below, we also show more explicitly how the water chemical data contributes highly valuable data for contextualizing the GHG dataset.

*Secondly, for the analysis of the water samples, the authors refer to a report by Vogt and Skancke, 2022 for the description of the analytical procedure and quality control. The report is freely available online (in Norwegian, which means this is not accessible to the international readership). From what I could see in that report, there is no mention of SiO2 measurement. If these data are included in the manuscript, the analytical procedure (instrument, detection limit, precision) for all the parameters measured on water samples including SiO2 should be included in the manuscript to be accessible to the reader. Regarding the measurements of dissolved silicon concentration (sometimes expressed as SiO2 in mg/l), the authors should refer to SiO2 concentration and avoid the use of the terminology silica. Because silica corresponds to the solid phase.*

We agree that a reference to a report in English is more appropriate. Hence, we changed the reference from Vogt and Skancke 2022 to Gundersen et al. 2025, which is openly available. This report contains all information on analytical procedures that are included in Vogt and Skancke, as well as information on $SiO_2$ procedures. Additionally, laboratory methods are available in the data repository (https://doi.org/10.4211/hs.41faf3d6c3f245259ea820740291789c).

The terminology used to refer to dissolved $SiO_2$ may vary across research fields, but in our field, it is common to refer to $SiO_2$ as silica. Sjøberg (1996) states that "silica ($SiO_2$) is regulated by (…) processes such as dissolution/precipitation" and notes that silica exists both in the solid phase (quartz) and in the dissolved phase. Additionally, the paper discusses "the solubility of silica," further supporting our usage. Given this established terminology, we will continue to refer to $SiO_2$ as silica in our manuscript.

*Thirdly, the method section refers to alkalinity. There is no information about how this was measured. The caption of Table 3 specifies that for this study DIC is considered as the sum of dissolved CO2 and bicarbonate. There is a need for more precision on the methodology and for the use of a similar terminology throughout the manuscript to avoid any confusion. And there is a mention for TOC analysis in mg/L. The TOC is usually used to refer to the total organic carbon concentration in a solid phase. What is the TOC for the water? This should be modified and redefined.*

We have removed alkalinity from the methods section, as it is not referenced elsewhere in the manuscript—thank you for catching this.

Regarding TOC, we recognize that terminology can vary across disciplines. In our study, TOC refers to the total organic carbon concentration in water, which includes both particulate and dissolved OC. This usage is well established in the literature (e.g., Visco et al., 2005), and we will continue to use TOC to describe the combined POC and DOC fractions in our water samples. L179 now reads: "Chemical analysis of pH, electrical conductivity (EC), and concentrations of (…) total organic carbon (TOC) (…)"

*Fourthly, the manuscript is based on a large dataset of water samples collected over two years, and median values are presented in Table 2, but a full access to the individual dataset should be made possible to review the manuscript.*

We fully agree on the importance of data transparency. We have ensured that the full dataset is accessible in a public repository, allowing for a thorough review of our results. The data are available through an online repository and have been added to the Data Availability section of the manuscript as well as cited (https://doi.org/10.4211/hs.41faf3d6c3f245259ea820740291789c)

*Comments by sections*

*Abstract*

*L18: the authors use the terminology of "organic acidity" to refer to "organic acids" as a driver for the pH. This should be revised, and the terminology "organic acidity" should be removed from the manuscript (L18, L20, L318, L497, L516).*

We acknowledge the importance of precise terminology in describing the role of DOC in influencing pH. However, we respectfully maintain that organic acidity is the appropriate term in this context as it describes the bulk acid-base properties of DOC rather than implying the presence of specific organic acids. DOC in natural waters consists of a complex mixture of organic molecules, with acidity arising from the dissociation of functional groups rather than discrete compounds alone. This term aligns with its use in the literature (e.g. Vogt et al., 2024) to describe DOC's role in pH regulation in aquatic systems. We hope this explanation clarifies our reasoning.

*L27-28: check that the 4 is as a subscript for CH4*

Thank you for catching this formatting error. We appreciate the attention to detail and have corrected the subscript for $CH_4$ accordingly.

*Introduction*

*L35-36, L40: the authors should incorporate recent publications with the permafrost carbon pool on land of 1460-1600 Pg C, and specify that they refer to the permafrost instead of northern latitude regions (Meredith et al 2019, IPCC Special Report on the Ocean and Cryosphere in a Changing Climate; Schuur et al 2022, https://doi.org/10.1146/annurev-environ-012220-011847; Strauss et al 2024, https://doi.org/10.1016/B978-0-323-99931-1.00164-1).*

We appreciate the suggestion to incorporate more recent publications on the permafrost carbon pool and acknowledge the importance of distinguishing between permafrost-specific regions and broader northern latitude areas. While permafrost in northern Norway is highly sporadic and limited, making a case for its inclusion within general northern latitude regions, we recognize the relevance of permafrost dynamics to our discussion. We propose to add the mentioned publications to the introduction, with the following text:

L35: "Northern latitude permafrost regions hold one of the largest terrestrial carbon reservoirs on the planet (Schuur et al., 2008; Schuur et al., 2015; Walter et al., 2006). Although covering only about 15% of global soils, permafrost regions contain an estimated 1400-1600 Pg of organic carbon (OC) (Hugelius et al., 2014; Schuur et al., 2022; Strauss et al., 2025), making these areas a critical component of the global carbon cycle."

L49-53: "While the large-scale thaw of permafrost is widely recognized (Leppiniemi et al., 2023) and permafrost regions warm 3-4 times faster than the global average (Meredith et al., 2019), the timing, magnitude, and pathways of carbon release remain uncertain, influenced by processes such as burial, mobilization, lateral export, and mineralization (Verdonen et al., 2023; Vonk et al., 2015)."

*L112: the motivation of the paper is the C cycling, and this is highly relevant. The justification for the need of the other parameters being measured on water should be clarified (also see major comments about the clarification needed for the analysis).*

Thank you for your comment. This concern was also raised by Reviewer 1, which highlights that we need to better justify the inclusion of water chemistry data and its relevance to understanding GHG processing. We acknowledge that our original manuscript did not sufficiently clarify the value of these measurements in addressing our research questions. To improve this, we propose a better articulation of the knowledge gaps that our study addresses and the role of water chemistry in understanding carbon cycling in degrading permafrost landscapes, in the introduction. Additionally, we have started the process of restructuring the Discussion section to begin with an analysis of hydrochemical differences across our study sites, emphasizing the distinct hydrological influences shaping water chemistry in thermokarst ponds and streams.

While carbon cycling remains the primary focus, we maintain that broader water chemistry parameters provide relevant context for understanding the environmental conditions influencing these processes. The water chemistry data allow us to distinguish between hydrologically different systems, such as the DOC-rich thermokarst ponds, which receive input from degrading permafrost, and streams, where groundwater contributions from catchment soils play a dominant role. This distinction is crucial for interpreting GHG dynamics, and we are incorporating clearer justifications as we revise the manuscript.

*Methods*

*L145, and L165: specific information about the number of water samples collected in each waterbody, the date, the time of the day (relative to the precipitation events) should be provided in details. For the data presentation, the number of data included (n = X) should be specified.*

Thank you for your suggestion. A table has been added to the supplementary material detailing the sampling dates, number of samples per site, and sample types. Additionally, this information, along with all measured parameters, is available in our public data repository. Since the manuscript does not address seasonal or daily variation, we maintain that specifying the exact timing of sampling is not necessary for the main text.

*L171-191: see earlier comment as a "major comment" about the water chemistry section*

We hope that we have addressed these concerns in our response to the earlier major comment regarding the water chemistry section.

*Results*

Comment on Results and Discussion: Thank you for your detailed feedback on the results and discussion sections. After reviewing your comments and those from RC1, we recognized the need for a clearer and more structured discussion of the water chemistry results and their significance. To address this, we are undertaking a revision of the discussion section:

- We have introduced a new section, 4.1 Hydrochemical contrasts between thermokarst ponds and wetland streams, which directly addresses the hydrochemical differences observed in our study. This section integrates the water chemistry results and provides a more structured interpretation of the data.
- The previous section 4.1 Thermokarst ponds as hotspots of methane emissions, has been revised and renamed 4.2, with improved contextualization now that we have an initial section discussing water chemistry. The reviewer correctly pointed out that the original section 4.2, Impact of thawing peat plateaus on water chemistry (L425-459), did not sufficiently link to the results of our study. In response, we have removed this section and integrated its key interpretations and context into the newly created discussion of water chemistry, ensuring a clearer connection between our findings and their broader implications.

- Interpretations that were previously embedded in the results section have been moved to the discussion now that there is a clear place for them when discussing the hydrochemical contrasts of the site, ensuring a clearer distinction between data presentation and interpretation.

These changes provide a stronger link between our results and the discussion, making it more relevant to the data we present while also engaging with existing literature.*L287-333: The first section of the results should be re-structured to present the data step by step, and to move the part related to the interpretation of the data to the discussion. This section should consider the major comment about the methods to define the parameters that will be described in this section and presented in Table 2 and in Figure 2.*

We hope this comment has been sufficiently answered by our response to the major comment in the methods as well as our Comment on Results and Discussion above going into some of the details about how we have made a new discussion section and transferring some results there.

*L291: "at the top" of what?*
*L293: "end of the curve" – which?*

These sentences have been revised for clarity: L291:

"The water bodies aligned along the inverse DOC-pH relationship with TK-Pond 3 exhibiting the highest DOC and lowest pH, followed by TK-Pond 2 and TK-Pond 1."

And L293: "The TK-Drain usually held an intermediate position between the thermokarst ponds and the wetland streams, which were found at the high pH – low DOC end of the DOC-pH relationship."

*L296: this is for the discussion section*

Please see our above Comment on Results and Discussion outlining our changes to the aforementioned sections.

*L299-300: The method section specifies that the totN data will not be presented for a reason, so there should be no mention of these data in the results and discussion*

Thank you for your comment. We understand the concern regarding the mention of total nitrogen (totN) data in the results section and we have removed the mention. We feel, however, it is important to justify the absence of totN data, as excluding it entirely might raise other questions. Therefore, we included a brief explanation in the methods section, stating that the totN data are available in the data repository but are not presented in the manuscript due to incomplete digestion. We believe this approach allows us to be transparent about the issue, while acknowledging the potential value of the data. To avoid confusion, we will not mention organic N in the results, but will clarify that these estimates, though potentially unreliable, are available in the data repository with a cautionary note about their limitations. We believe this balances being transparent about the data's status while not presenting unreliable information as definitive in the text.

*L306: "somewhat higher" should be more specific with statistics*

We agree that specifying the statistics would improve the clarity of this statement. The following change was made:

"The DOM quality indicator, sUVa, a proxy for aromaticity, was slightly higher in the wetland streams than in the thermokarst ponds, although the difference was not significant."

*In Table 2: NO4 should be NO3*

Thank you for finding this mistake. "$NO_4$" has been corrected to "$NO_3^-$" in Table 2.

*L319-321: this is for the discussion section*

Please refer to our Comment on Results and Discussion above outlining our changes.

*Figure 3: specify in the caption that this is dissolved concentration in CO2 and CH4 in waters. There is no description of the lower panels.*

We appreciate the suggestion to clarify the caption for Figure 3. While we had focused on describing the upper panels, we now see that the lower panels were overlooked. To address this, we have revised the caption to explicitly state that the figure presents dissolved concentrations of $CO_2$ and $CH_4$ in the waters and will include a clear description of the lower panels. Specifically:

"Variations in dissolved $CO_2$ (panels a–c) and $CH_4$ (panels d–f) across sampling sites in relation to pH, DOC and totP."

*Discussion*

*L425-459: in the second section of the discussion, there is not link to the data of the present study. The discussion should compare the original data and discuss them with the literature. Also the title of the section should specify what part of the water chemistry will be discussed in the section.*

Please see our Comment on Results and Discussion above where we address this comment and the changes to the section.

*Section 4.3: references to the data presented in the results should be included by adding a link to a figure or a table at L461, L468, L475.*

We have revised Section 4.3 to include references to the data presented in the Results by adding direct reference to the relevant figures and tables at L461, L468, and L475, as suggested. This will ensure the discussion is more directly tied to the results.

*L511: variability of what?*

Thank you pointing out the imprecision of this sentence. To clarify, the "variability" refers to the spatiotemporal variability of ecological and hydrological factors, particularly during thaw cycles, which can influence GHG emissions and carbon feedbacks in permafrost landscapes. The passage has been revised as follows:

"These findings reflect the complex interplay of ecological and hydrological factors shaping GHG emissions in permafrost landscapes. Turetsky et al. (2020) and Pirk et al. (2024) both emphasized the need for further research on the spatiotemporal variability of these factors, particularly during thaw cycles, as shifts in hydrological connectivity, OM transport, and microbial activity can significantly influence the GHG emissions and permafrost-C feedbacks. Improving our understanding of these dynamics is essential for refining predictions of permafrost-C feedbacks in a changing climate."

*L538-540: Data availability: the manuscript is based on a large dataset of water samples collected over two years, and median values are presented in Table 2, but a full access to the individual dataset should be made possible to review the manuscript.*

We fully agree on the importance of data transparency. As we addressed in response to the similar major comment, we have made the full dataset available in a public repository to ensure thorough review of our results.

**References**:

Sjöberg, S.: Silica in aqueous environments, J. Non-Cryst. Solids, 196, 51–57, https://doi.org/10.1016/0022-3093(95)00562-5, 1996.

Visco, G., Campanella, L., and Nobili, V.: Organic carbons and TOC in waters: an overview of the international norm for its measurements, Microchem. J., 79, 185–191, https://doi.org/10.1016/j.microc.2004.10.018, 2005.

Vogt, R. D., Garmo, Ø. A., Austnes, K., Kaste, Ø., Haaland, S., Sample, J. E., Thrane, J.-E., Skancke, L. B., Gundersen, C. B., and de Wit, H. A.: Factors Governing Site and Charge Density of Dissolved Natural Organic Matter, *Water*, 16(12), 1716, https://doi.org/10.3390/w16121716, 2024.

---

## Author Response (AR1)

We would like to thank the editor and reviewers for their thoughtful and constructive comments on our manuscript. We greatly appreciate the time and effort that went into reviewing our work and for the helpful suggestions that have significantly improved the quality of the manuscript.

In the following pages, we provide a detailed, point-by-point response to each of the comments raised. For clarity, the reviewers' comments are reproduced in *blue italicized text*, followed by our responses and a description of the changes made in the revised manuscript.

We thank you again for the opportunity to revise and resubmit our work.

Sincerely,

Jacqueline Knutson

Corresponding author on behalf of all co-authors

*One major issue I would like to bring up and that the authors do not mention is the role of the redox scale in explaining the patterns here. The authors have a large gradient in SO4, linked to pH and DOC, that is clearly connected to the low CH4 concentrations in the stream, and a more clear exploration of this resolution would be a nice contribution of the paper. If SO4 is abundant then is not energetically favorable to produce methane, so this could explain a lot the patterns regardless of the system.*

We acknowledge the importance of redox conditions for GHG production as outlined in L72 in the introduction and L418-424 and L430 in the Discussion. In our study, the ponds were always oxic – this is mentioned in L341-342 (see also Table 3). In the presence of oxygen ($O_2$), the role of $SO_4$ as electron acceptor in freshwaters is negligible (e.g., Clayer et al. 2016).

However, in the sediments there were likely anoxic conditions promoting $CH_4$ production. Therefore, the $CH_4$ found in the ponds and in the streams likely originates from anoxic sediments (e.g., L418-424) and $CH_4$ production in oxic pond and stream waters is likely inhibited, not by $SO_4$, but by high $O_2$ concentrations. To highlight the role of redox conditions for GHG production and $CH_4$ production inhibition by $O_2$, we will add the following sentences L419:
"Under high $O_2$ concentration, $CH_4$ production is typically inhibited (Clayer et al., 2016). Hence the high $CH_4$ concentration in the thermokarst ponds (5000 times oversaturation) and, to a lesser degree in the wetland streams (30 times oversaturation), is likely sustained by $CH_4$ efflux from anoxic sediments."

Please note that the $CH_4$ concentration in the stream is not "low", as stated by the reviewer. While it is lower than in the ponds it is still 30 times higher than atmospheric saturation; see L344-347.

*The manuscript « Water chemistry and greenhouse gas concentrations in waterbodies of a thawing permafrost peatland complex in northern Norway » presents original data of greenhouse gas (GHG) concentrations in water bodies from a peatland area in the permafrost region in Norway collected over 2 years of monitoring. The dataset for the GHG is highly valuable for the community given the little knowledge on these emerging environments. The data are accompanied with water chemical composition, but some additional information is needed before these additional data can be used in the manuscript (see comments below). I would really encourage the authors to address the comments below to allow the publication of this highly valuable GHG dataset.*

*Major comments*

*The most important comments are about the water chemistry measurements (sampling, analytical procedures, and data availability).*

*Firstly, according to the method description, the unfiltered waters were sampled in HDPE bottles and kept at 4°C until analysis. The method does not specify whether filtration was performed. For the analysis of SiO2, SO4, NO3, NH4, totP concentrations, if filtration was performed, it should be specified with what type of filter (material), what pore size, and how long after sampling. For DOC and POC analysis, the information about filtration should be more specific and specify which type of filter, pore size, and how long after sampling. Importantly, for the analysis of SiO2, SO4, NO3, NH4, totP concentrations, if no filtration was performed, I would strongly recommend not to use the data, and to focus the manuscript on the C cycling. Unfiltered waters may contain small sized particles which continue to interact with the dissolved phase after sampling (dissolution, adsorption, etc). This is why it is conventionally recommended for this type of water analysis to filter within 24h to 48h.*

We would like to respond to the major comment raised regarding the place of the water chemical data in the manuscript. We applied generally accepted methods for sampling and analyzing freshwaters in this manuscript but may not have described these methods in sufficient detail, for which we stand corrected. To assess whether delayed filtration influenced our results, we conducted a filtration comparison experiment in September 2022. This test showed no significant differences between field-filtered and lab-filtered samples for DOC, pH, $SiO_2$, $SO_4$, TOC, TOTN, or conductivity. Minor differences were observed for $PO_4$-P and total P in TK-Pond 1 and TK-Pond 3, likely due to particle flocculation between collection and filtration. A more notable difference was found in $NH_4$-N at TK-Pond 1, suggesting some variability in ammonium concentration. However, overall, the results indicate that our sampling and filtration approach provided reliable data. We found that this justified our sampling methods, e.g. sample water in HDPE bottles, keep them cool (4°C) and dark, transport them as quickly as possible to the lab and do filtration for some parameters in the laboratory. The field filtration that the reviewer suggests would have taken several hours extra per field day, which would have limited other data collection. Given these findings, we maintain that our water chemistry dataset is scientifically robust and suitable for inclusion in the manuscript. To improve clarity and transparency, we have updated the methods section with more detailed information on laboratory procedures and have added a full description of the filtration comparison experiment and its results in the supplementary material. Below, we also show more explicitly how the water chemical data contributes highly valuable data for contextualizing the GHG dataset.

*Secondly, for the analysis of the water samples, the authors refer to a report by Vogt and Skancke, 2022 for the description of the analytical procedure and quality control. The report is freely available online (in Norwegian, which means this is not accessible to the international readership). From what I could see in that report, there is no mention of SiO2 measurement. If these data are included in the manuscript, the analytical procedure (instrument, detection limit, precision) for all the parameters measured on water samples including SiO2 should be included in the manuscript to be accessible to the reader. Regarding the measurements of dissolved silicon concentration (sometimes expressed as SiO2 in mg/l), the authors should refer to SiO2 concentration and avoid the use of the terminology silica. Because silica corresponds to the solid phase.*

We agree that a reference to a report in English is more appropriate. Hence, we changed the reference from Vogt and Skancke 2022 to Gundersen et al. 2025, which is openly available. This report contains all information on analytical procedures that are included in Vogt and Skancke, as well as information on $SiO_2$ procedures. Additionally, laboratory methods are available in the data repository (https://doi.org/10.4211/hs.41faf3d6c3f245259ea820740291789c).

The terminology used to refer to dissolved $SiO_2$ may vary across research fields, but in our field, it is common to refer to $SiO_2$ as silica. Sjøberg (1996) states that "silica ($SiO_2$) is regulated by (…) processes such as dissolution/precipitation" and notes that silica exists both in the solid phase (quartz) and in the dissolved phase. Additionally, the paper discusses "the solubility of silica," further supporting our usage. Given this established terminology, we will continue to refer to $SiO_2$ as silica in our manuscript.

*Thirdly, the method section refers to alkalinity. There is no information about how this was measured. The caption of Table 3 specifies that for this study DIC is considered as the sum of dissolved CO2 and bicarbonate. There is a need for more precision on the methodology and for the use of a similar terminology throughout the manuscript to avoid any confusion. And there is a mention for TOC analysis in mg/L. The TOC is usually used to refer to the total organic carbon concentration in a solid phase. What is the TOC for the water? This should be modified and redefined.*

We have removed alkalinity from the methods section, as it is not referenced elsewhere in the manuscript—thank you for catching this.

Regarding TOC, we recognize that terminology can vary across disciplines. In our study, TOC refers to the total organic carbon concentration in water, which includes both particulate and dissolved OC. This usage is well established in the literature (e.g., Visco et al., 2005), and we will continue to use TOC to describe the combined POC and DOC fractions in our water samples. L179 now reads: "Chemical analysis of pH, electrical conductivity (EC), and concentrations of (…) total organic carbon (TOC) (…)"

*Fourthly, the manuscript is based on a large dataset of water samples collected over two years, and median values are presented in Table 2, but a full access to the individual dataset should be made possible to review the manuscript.*

We fully agree on the importance of data transparency. We have ensured that the full dataset is accessible in a public repository, allowing for a thorough review of our results. The data are available through an online repository and have been added to the Data Availability section of the manuscript as well as cited (https://doi.org/10.4211/hs.41faf3d6c3f245259ea820740291789c)

MINOR COMMENTS

*L15: The second sentence starting "at the Iskoras site in Northern Norway..." feels a bit rushed to introduce the methods part in the abstract, presenting the knowledge gap or question before would help the reading.*
We revised the abstract with the following addition to better introduce the knowledge gap at L15: "Transitions from isolated thermokarst ponds to interconnected wetlands can substantially alter GHG dynamics, but the biogeochemical processes underlying these shifts are not well understood— particularly in the sporadic permafrost zones of Fennoscandia, where such small systems are infrequently reported. To address this, we investigated water chemistry, dissolved organic matter (DOM) processing, and GHG fluxes over two years at the Iškoras site in northern Norway, where a degrading peat plateau includes both thermokarst ponds and an adjacent wetland stream."

*L18: the authors use the terminology of "organic acidity" to refer to "organic acids" as a driver for the pH. This should be revised, and the terminology "organic acidity" should be removed from the manuscript (L18, L20, L318, L497, L516).*

We acknowledge the importance of precise terminology in describing the role of DOC in influencing pH. However, we respectfully maintain that organic acidity is the appropriate term in this context as it describes the bulk acid-base properties of DOC rather than implying the presence of specific organic acids. DOC in natural waters consists of a complex mixture of organic molecules, with acidity arising from the dissociation of functional groups rather than discrete compounds alone. This term aligns with its use in the literature (e.g. Vogt et al., 2024) to describe DOC's role in pH regulation in aquatic systems. We hope this explanation clarifies our reasoning.

*L24: It is maybe better to keep the abstract with one paragraph.*
The abstract is one paragraph and a connecting sentence has been added the L24 now reads:
"…ponds were primarily linked to DOM mineralization. Despite differences in chemistry, DOM mineralization…"

*L27-28: check that the 4 is as a subscript for CH4*

Thank you for catching this formatting error. We appreciate the attention to detail and have corrected the subscript for $CH_4$ accordingly.

*L35-36, L40: the authors should incorporate recent publications with the permafrost carbon pool on land of 1460-1600 Pg C, and specify that they refer to the permafrost instead of northern latitude regions (Meredith et al 2019, IPCC Special Report on the Ocean and Cryosphere in a Changing Climate; Schuur et al 2022, https://doi.org/10.1146/annurev-environ-012220-011847; Strauss et al 2024, https://doi.org/10.1016/B978-0-323-99931-1.00164-1).*

We appreciate the suggestion to incorporate more recent publications on the permafrost carbon pool and acknowledge the importance of distinguishing between permafrost-specific regions and broader northern latitude areas. While permafrost in northern Norway is highly sporadic and limited, making a case for its inclusion within general northern latitude regions, we recognize the relevance of permafrost dynamics to our discussion. We propose to add the mentioned publications to the introduction, with the following text:

L35: "Northern latitude permafrost regions hold one of the largest terrestrial carbon reservoirs on the planet (Schuur et al., 2008; Schuur et al., 2015; Walter et al., 2006). Although covering only about 15% of global soils, these regions contain an estimated 1400-1600 Pg of organic carbon (OC) (Hugelius et al., 2014; Schuur et al., 2022; Strauss et al., 2025), making them a critical component of the global carbon cycle."

L49-53: "While the large-scale thaw of permafrost is widely recognized (Leppiniemi et al., 2023) and permafrost regions warm 3-4 times faster than the global average (Meredith et al., 2019), the timing, magnitude, and pathways of carbon release remain uncertain, influenced by processes such as burial, mobilization, lateral export, and mineralization (Verdonen et al., 2023; Vonk et al., 2015)."

*L53: the formation "of" thermokarst ponds. Missing an "of".*
We thank the reviewer for pointing out this omission and L53 now reads:

"As these features degrade, permafrost thaw is often abrupt and subsidence and collapse is to be expected, leading to the formation of thermokarst ponds, as excess ground ice is lost (Martin et al., 2021).".

*L83: The introduction is overall good, but there is an abuse of "these" and "their" (10 in this page),*

*which makes the reading a bit hard with the indirect references. Having a more direct writing would help the reader, and some of them are unknown what they refer to.*

We have revisited the introduction and agree with your suggestion. The introduction has been revised to minimize indirect references and make the writing clearer, specifically in

L62: "While the processes driving  permafrost thaw and landscape transformations, such as thermal disturbances, vegetation…"

L73: "Despite  the small size of thermokarst ponds, these waterbodies can contribute significantly to regional C fluxes, with CH4 and CO2 supersaturation levels often surpassing those of larger lakes—whether thermokarst or not —or…"

L76: "However,  the contributions of thermokarst ponds are often overlooked in large-scale C assessments…"

L90: "Understanding the dynamics of these evolving permafrost and wetland systems is critical for assessing…"

L93: "provides a valuable opportunity to investigate carbon cycling in rapidly evolving subarctic landscapes …"

*L99: I would make those critical questions more explicit.*

We agree and made the following change starting at L96:

"While existing studies emphasize the importance of quantifying $CH_4$ and $CO_2$ fluxes in these environments and their implications for C budgets (Abnizova et al., 2012; Matveev et al., 2018), the interactions between hydrology, vegetation and carbon processing are not well understood. Yet such processes are central to key questions regarding how transitions between permafrost, thermokarst, and wetland systems influence C dynamics, and whether these landscapes function as net C sources or sinks under changing climatic conditions (Sim et al., 2021). In particular, peatland ponds and thermokarst waterbodies exhibit unique biogeochemical cycling from lakes, driven more by internal dynamics than external watershed inputs (Arsenault et al., 2022). These differences remain poorly represented in both observational datasets and Earth system models."

*L102: And the gaps are also not specified. This should be way more explicit.*

Following the changes outlining the critical questions, we more clearly state our study's place starting at L104:

"Over two years, we collected a novel dataset combining biogeochemical measurements with C flux data from thermokarst ponds and a wetland stream within a small permafrost peatland plateau undergoing rapid permafrost degradation. This setting captures a landscape in active transition from isolated thermokarst ponds to interconnected wetlands."

*L109: Where is this OM mobilized from? Permafrost thaw? Recent GPP?*

The OM is mobilized from permafrost thaw, and this is now added to the text at L109:

"…and (3) recently mobilized OM from thawing permafrost presents a labile source of C promoting $CO_2$ production in thermokarst water bodies compared to wetland streams."

*L112: the motivation of the paper is the C cycling, and this is highly relevant. The justification for the need of the other parameters being measured on water should be clarified (also see major comments about the clarification needed for the analysis).*

While carbon cycling remains the primary focus, we maintain that broader water chemistry parameters provide relevant context for understanding the environmental conditions influencing these processes. The water chemistry data allow us to distinguish between hydrologically different systems, such as the DOC-rich thermokarst ponds, which receive input from degrading permafrost, and streams, where groundwater contributions from catchment soils play a dominant role. This distinction is crucial for interpreting GHG dynamics, and we have increased our discussion section to address this comment as well. See also the changes to L96 and L104 that now address this comment.

*L126: What does the +/- refer to here?*
As we do not report the standard deviation when reporting the temperature and precipitation during the study period in the main text, we have removed this for the normal period from the main text as well. A clarification that values represent mean ± standard deviation/interannual variation has been added to the description of Table 1 where these numbers are reported for both the normal period and the study period.

*L145, and L165: specific information about the number of water samples collected in each waterbody, the date, the time of the day (relative to the precipitation events) should be provided in details. For the data presentation, the number of data included (n = X) should be specified.*

A table has been added to the supplementary material detailing the sampling dates, number of samples per site, and sample types. Additionally, this information, along with all measured parameters, is available in our public data repository. Since the manuscript does not address seasonal or daily variation, we maintain that specifying the exact timing of sampling is not necessary for the main text. L145 now reads: "Measurements and samples were taken approximately monthly in the ice-free season from May until October in 2021 and 2022 (full details in Table SI 1)."

*L165: Would be helpful to specify what those measurements are for, at least for "dark incubations".*
The dark incubations estimate greenhouse gas production rates and the sentence at L165 now reads:

"…dissolved gases, $CO_2$ emissions, GHG production using the dark incubation method, and high-frequency monitoring of water height and temperature."

*L171-191: see earlier comment as a "major comment" about the water chemistry section*

A full overview of our methods and field filtration and preservation experiment is now included in the supplementary materials. In addition, changes were made to be clearer in this section by changing the language at L176:

"…The samples were then transported by car and plane back to the laboratory and delivered for chemical analysis where they were kept at 4ºC until analysis. To ensure that our sampling procedures were suitable for the determination of nutrient concentrations, we performed a comparison of different field procedures including acidification and filtration in the field (see Supplementary Methods). We found no significant differences between procedures (Table SI 4). Chemical analysis of pH, electrical conductivity (EC), and concentrations of sulfate ($SO_4^{2-}$), silica ($SiO_2$), ammonium ($NH_4^+$), nitrate ($NO_3^-$), total phosphorous (totP), total organic carbon (TOC), DOC and particulate organic carbon (POC) in the water samples was performed at accredited laboratories at the Norwegian Institute for Water Research (NIVA); methods for analysis and quality control are described in the ICP Waters Programme Manual (Gunderson et al., 2025). The samples were not fully digested according to standard procedures required for the determination of total nitrogen (totN), hence totN values are expected to be underestimated and are therefore not shown in the manuscript, although the values were enough to confirm the dominant form of N was organic (Thrane et al., 2020)."

*L178: The ions need their plus and minus*
The ions have gotten their pluses and minuses throughout the manuscript and in the figures.

*L197: The calculations for CO2 are very unclear. If samples are acidified, all DIC becomes CO2 before the shaking. Was this done for all samples or one set for DIC (acidified) and the other for CO2 (not acidified)?*
Thank you for your comment. The duplicate samples followed the same procedure in that all DIC was converted to $CO_2$ by acidification. To avoid confusion, we rephrased L196:

"Duplicate gas samples were collected according to Valiente et al. (2022) with 50 mL syringes. These were filled and sealed underwater without air bubbles to prevent gas loss."

L198: "All samples were acidified with 0.6 mL of 3% HCl to achieve pH <2, ensuring DIC was present as $CO_2$."

Note that calculations are given, including back calculations of $CO_2$ from DIC, in L212-221. We have changed L216:

"The bicarbonate concentrations were calculated based on pH, total dissolved $CO_2$ (after acidification), and the temperature-adjusted first dissociation constant ($pK_1$ = 6.41 at 25°C; Stumm and Morgan (2013)) of the carbonic acid equilibrium."

*L200: write 15 instead of text.*
L200 was rewritten to prevent beginning a sentence with a number: "repeated thrice and then 15 mL of headspace gas was transferred to 12 mL evacuated vials, and water temperature in the syringe was measured immediately after gas transfer."

*L240: Change "processing" to production.*
We appreciate the suggestion to use "production" instead of "processing" and have edited L240.

Comment on Results and Discussion: After considering comments from both reviewers regarding the presentation of the results and discussion sections, we recognized the need for a clearer and more structured discussion of the water chemistry results and their significance. To address this, we have fully revised the discussion section:
1. We have introduced a new section, 4.1 which directly addresses the hydrochemical differences observed in our study. This section integrates the water chemistry results and provides a more structured interpretation of the data. For example, to respond to the comment by RC1 at L466, we aimed to strengthen our interpretation that groundwater inputs, associated with mineral weathering is influencing $CO_2$ emissions in the streams by explicitly discussing our results and the role of weathering at the start of the discussion and bringing in more relevant references (e.g. Lehmann et al., 2023).
2. The previous section 4.1 Thermokarst ponds as hotspots of methane emissions, has been revised and renamed 4.2, with improved contextualization now that we have an initial section discussing water chemistry. RC2 correctly pointed out that the original section 4.2, Impact of thawing peat plateaus on water chemistry (L425-459), did not sufficiently link to the results of our study. In response, we have removed this section and integrated its key interpretations

and context into the newly created discussion of water chemistry, ensuring a clearer connection between our findings and their broader implications.

3. Interpretations that were previously embedded in the results section have been moved to the discussion now that there is a clear place for them when discussing the hydrochemical contrasts of the site, ensuring a clearer distinction between data presentation and interpretation.

These changes provide a stronger link between our results and the discussion, making it more relevant to the data we present while also engaging with existing literature.

*L287-333: The first section of the results should be re-structured to present the data step by step, and to move the part related to the interpretation of the data to the discussion. This section should consider the major comment about the methods to define the parameters that will be described in this section and presented in Table 2 and in Figure 2.*

We hope this comment has been sufficiently answered by our response to the major comment in the methods as well as our Comment on Results and Discussion above going into some of the details about how we have made a new discussion section and transferring some results there.

*L291: "at the top" of what?*
*L293: "end of the curve" – which?*

These sentences have been revised for clarity: L291:

"The water bodies aligned along the inverse DOC-pH relationship with TK-Pond 3 exhibiting the highest DOC and lowest pH, followed by TK-Pond 2 and TK-Pond 1."

And L293: "The TK-Drain usually held an intermediate position between the thermokarst ponds and the wetland streams, which were found at the high pH – low DOC end of the DOC-pH relationship."

*L296: this is for the discussion section*

L296 has been adjusted to read "Particulate OC concentrations were significantly higher and more variable in thermokarst ponds (1.2–3.4 mg L-1) compared to wetland streams (0.4–0.6 mg L-1), with greater variability observed at the Outlets than Inlet (Table 2)." And the interpretation has been moved to the discussion.

*L299-300: The method section specifies that the totN data will not be presented for a reason, so there should be no mention of these data in the results and discussion*

We understand the concern regarding the mention of total nitrogen (totN) data in the results section and we have removed the mention. We feel, however, it is important to justify the absence of totN data, as excluding it entirely might raise other questions. Therefore, we included a brief explanation in the methods section, stating that the totN data are available in the data repository but are not presented in the manuscript due to incomplete digestion in L186. We believe this approach allows us to be transparent about the issue, while acknowledging the potential value of the data. To avoid confusion, we will not mention organic N in the results, but will clarify that these estimates, though potentially unreliable, are available in the data repository with a cautionary note about their limitations. We believe this balances being transparent about the data's status while not presenting unreliable information as definitive in the text.

*L303: "SAR" is introduced here for the first time, this needs details in the methods.*
We currently introduce SAR in the methods section in L186-191. To be clearer, L190 now reads: "The absorbance values at 254 nm ($A_{\lambda 254nm}$) were used to calculate specific UV absorbance, expressed as

sUVa= $A_{\lambda 254nm}$/mg C L$^{-1}$, and the specific UV absorption ratio (SAR = $A_{\lambda 254nm}$/$A_{\lambda 400nm}$) was calculated for each sample."

*L306: "somewhat higher" should be more specific with statistics*

We agree that specifying the statistics improves this statement. The following change was made at L306:

"The DOM quality indicator, sUVa, a proxy for aromaticity, was slightly higher in the wetland streams than in the thermokarst ponds, although the difference was not significant."

*In Table 2: NO4 should be NO3*

Thank you for finding this mistake. "$NO_4$" has been corrected to "$NO_3^-$" in Table 2.

*L309: conductivity is commonly expressed as microsiemens/cm, so this would be a better unit.*
We have reported conductivity in mS m$^{-1}$, which aligns with ISO standards (such as ISO 7888:1985 for water quality—determination of electrical conductivity) and European scientific conventions. While mS cm$^{-1}$ is also used in some contexts, mS m$^{-1}$ is the preferred unit in European hydrogeology, water quality, and engineering disciplines.

*L319-321: this is for the discussion section*

Please refer to our Comment on Results and Discussion above outlining our changes.

*Figure 3: specify in the caption that this is dissolved concentration in CO2 and CH4 in waters. There is no description of the lower panels.*

While we had focused on describing the upper panels, we now see that the lower panels were overlooked. To address this, we have revised the caption to explicitly state that the figure presents dissolved concentrations of $CO_2$ and $CH_4$ in the waters and will include a clear description of the lower panels. Specifically:

"Variations in dissolved $CO_2$ (panels a–c) and $CH_4$ (panels d–f) across sampling sites in relation to pH, DOC and totP."

*L366: Here the DIC and CO2 plot are repeated, one should be changed.*
L366 has a new plot. While the DIC and $CO_2$ plots look repeated for the ponds, they are slightly different. This figure supports the importance of pH in carbonate chemistry and that in the ponds DIC and $CO_2$ are very similar because of the low pH. However, we understand that it looks like it has been repeated and edited the plots to show only $CO_2$. The plots for DIC is referred to in the text and can be found in the supplement as Figure SI 2.

*L422: It is a bit confusing to say "fresh OM input from themokarst development", as is probably "old carbon. Same in line 434.*
We appreciate this comment on the distinction between "fresh" and "old" carbon input. We have updated this to clarify that it refers to old carbon that is freshly available:
"The microbial activity responsible for CH4 production may be enhanced by the release of previously frozen OM newly available from ongoing thermokarst development (Crevecoeur et al., 2017), as recently observed in laboratory incubations with recently inundated peat material from the Iškoras site (Kjær, 2024)."

And
"…emit large amounts of CH4, particularly where old OC becomes available from peat decomposition from degrading palsa edges."

*L425-459: in the second section of the discussion, there is not link to the data of the present study. The discussion should compare the original data and discuss them with the literature. Also the title of the section should specify what part of the water chemistry will be discussed in the section.*

Please see our Comment on Results and Discussion above, specifically point 2, where we address this comment and the changes to the section.

*L448: What do you mean CO2 is sequestered here? May need a bit more explanation.*
Please refer to our Comment on Results and Discussion above outlining our changes.

*L466[sic]: I am not sure there is support for the carbonate inputs sustaining CO2 from streams here.*

To strengthen our interpretation that groundwater inputs, associated with mineral weathering is influencing $CO_2$ emissions in the streams, we have re-arranged the discussion showing more explicitly the role of weathering and the importance of water chemistry at the start of the discussion in a new section 4.1 as explained in our Comment on Results and Discussion above. L466 now reads: "…In contrast, streams exhibit enhanced $CO_2$ fluxes likely due to high turbulence and carbonate inputs from groundwater as described in section 4.1 and consistent with high dissolved cations (Table SI3). While Zolkos et al. (2018) highlight that carbonate weathering in thermokarst-affected streams can be driven from acidity from thermokarst ponds, our results suggest that the elevated cation concentrations are likely a reflection of groundwater influence, which can contribute both bicarbonate and dissolved $CO_2$ to sustain stream $CO_2$ fluxes (Lehmann et al., 2023; Duvert et al., 2018; Winterdahl et al., 2016)…".

*Section 4.3: references to the data presented in the results should be included by adding a link to a figure or a table at L461, L468, L475.*

We have revised Section 4.3 to include references to the data presented in the Results by adding direct reference to the relevant figures and tables at L461, L468, and L475, as suggested. This will ensure the discussion is more directly tied to the results

*L471: same here, groundwater can be rich of bicarbonates but also directly CO2, which is highly enriched in the groundwater. I would leave the discussion more broadly defined with DIC instead of getting into the weeds of the carbonate equilibrium here. A missing reference would be some more stream continuum focused, such as Hotchkiss et al 2014 (Nature Geoscience).*
L471 has been updated with new references and now reads: "This flux is consistent with the combined influence of turbulent flow and groundwater inputs enriched in DIC, consistent with earlier findings of mineral weathering contributions to DIC at Iškorasfjellet (Lehmann et al., 2023). These findings complement observations from other regions, where turbulence and bicarbonate supply are key drivers of CO2 release in streams (Lundin et al., 2013; Raymond et al., 2013). Streams likely benefit from continuous replenishment of CO2 from bicarbonates, which account for 60–70% of the DIC pool in boreal to Arctic streams and rivers (Wallin et al., 2018; Zolkos and Tank, 2020)…"

*L496: Same here, the mention of weathering is a bit off without data on that.*

Please refer to our Comment on Results and Discussion above where we discuss our results and weathering specifically in section 4.1. L496 now reads: "…driven by the combined effects of turbulent flow, groundwater-derived DIC, and mineral weathering inputs, whereas lower emissions from ponds are shaped by organic acidity, limited hydrological connectivity, and limited surface exchange. These dynamics…" and ties back to data presented in section 4.1.

*L511: variability of what?*

To clarify, the "variability" refers to the spatiotemporal variability of ecological and hydrological factors, particularly during thaw cycles, which can influence GHG emissions and carbon feedbacks in permafrost landscapes. L511 now reads:

"These findings reflect the complex interplay of ecological and hydrological factors shaping GHG emissions in permafrost landscapes. Turetsky et al. (2020) and Pirk et al. (2024) both emphasized the need for further research on the spatiotemporal variability of these factors, particularly during thaw cycles, as shifts in hydrological connectivity, OM transport, and microbial activity can significantly influence the GHG emissions and permafrost-C feedbacks. Improving our understanding of these dynamics is essential for refining predictions of permafrost-C feedbacks in a changing climate."

*L538-540: Data availability: the manuscript is based on a large dataset of water samples collected over two years, and median values are presented in Table 2, but a full access to the individual dataset should be made possible to review the manuscript.*

We fully agree on the importance of data transparency. As we addressed in response to the similar major comment, we have made the full dataset available in a public repository to ensure thorough review of our results.

**References**:

Sjöberg, S.: Silica in aqueous environments, J. Non-Cryst. Solids, 196, 51–57, https://doi.org/10.1016/0022-3093(95)00562-5, 1996.

Visco, G., Campanella, L., and Nobili, V.: Organic carbons and TOC in waters: an overview of the international norm for its measurements, Microchem. J., 79, 185–191, https://doi.org/10.1016/j.microc.2004.10.018, 2005.

Vogt, R. D., Garmo, Ø. A., Austnes, K., Kaste, Ø., Haaland, S., Sample, J. E., Thrane, J.-E., Skancke, L. B., Gundersen, C. B., and de Wit, H. A.: Factors Governing Site and Charge Density of Dissolved Natural Organic Matter, *Water*, 16(12), 1716, https://doi.org/10.3390/w16121716, 2024.